**Data Availability Statement:** All relevant data are within the manuscript Raw data are held in the

# Neuroinflammation, body temperature and behavioural changes in CD1 male mice undergoing acute restraint stress: An exploratory study

Veronica Redaelli[1], Alice Bosi[2], Fabio Luzi[1], Paolo Cappella[3], Pietro Zerbi[4], Nicola Ludwig[5], Daniele Di Lernia[6], John Vincent Roughan[7], Luca Porcu[8], Davide Soranna[9], Gianfranco Parati[2,10], Laura Calvillo[2]*

1 Department of Biomedical, Surgical and Dental Sciences–One Health Unit, Università degli Studi di Milano, Milan, Italy, 2 Department of Cardiovascular, Neural and Metabolic Sciences, Istituto Auxologico Italiano, IRCCS, San Luca Hospital, Milan, Italy, 3 Research & Services Dept, E.m.c2 srl, Varese, Italy, 4 Dipartimento di Scienze Biomediche e Cliniche "L. Sacco", Università degli Studi di Milano, Milan, Italy, 5 Dipartimento di Fisica, Università degli Studi di Milano, Milan, Italy, 6 Humane Technology Lab, Dipartimento di psicologia, Università Cattolica del Sacro Cuore, Milan, Italy, 7 Institute of Neuroscience, Comparative Biology Centre, Newcastle University, Newcastle upon Tyne, United Kingdom, 8 Laboratory of Methodology for Clinical Research, Oncology Department, Istituto di Ricerche Farmacologiche Mario Negri IRCCS, Milan, Italy, 9 Biostatistics Unit, Istituto Auxologico Italiano, IRCCS, Milan, Italy, 10 Department of Medicine and Surgery, University of Milano-Bicocca, Milan, Italy

* l.calvillo@auxologico.it

## Abstract

### Background

Animal models used to study pathologies requiring rehabilitation therapy, such as cardio-vascular and neurologic disorders or oncologic disease, must be as refined and translation-ally relevant as possible. Sometimes, however, experimental procedures such as those involving restraint may generate undesired effects which may act as a source of bias. However, the extent to which potentially confounding effects derive from such routine procedures is currently unknown. Our study was therefore aimed at exploring possible undesirable effects of acute restraint stress, whereby animals were exposed to a brightly lit enclosed chamber (R&L) similar to those that are commonly used for substance injection. We hypothesised that this would induce a range of unwanted physiological alterations [such as neuroinflammatory response and changes in body weight and in brown adipose tissue (BAT)] and behavioural modification, and that these might be mitigated via the use of non-aversive handling methods: Tunnel Handling (NAH-T) and Mechanoceptive Handling (NAH-M)) as compared to standard Tail Handling (TH).

### Methods

Two indicators of physiological alterations and three potentially stress sensitive behavioural parameters were assessed. Physiological alterations were recorded via body weight changes and assessing the temperature of Brown Adipose Tissue (BAT) using infra-red

public repository http://doi.org/10.5281/zenodo.4396025.

**Funding:** This publication was made possible by funds of the Italian Ministry of Health and by grant number NC/S000887/1 provided to Dr Johnny Roughan by the UK NC3R's. The sponsors or funders did not play any role in the study design, data collection and analysis, decision to publish, or preparation of the manuscript. Paolo Cappella is a full-time employee in E.M.C2 S.r.l. and has supported in vivo laboratory activities in accordance with authors requests. The Company has no commercial interests in the therapeutic area of this study and did not play any role in the study design, providing only financial support in the form of author Paolo Cappella' salary.

**Competing interests:** Authors declare that there are no conflicts of interest. Paolo Cappella is a full-time employee in E.M.C2 S.r.l. and has supported in vivo laboratory activities in accordance with authors requests. The Company has no commercial interests in the therapeutic area of this study and did not play any role in the study design, providing only financial support in the form of author Paolo Cappella' salary.

thermography (IRT), and at the end of the experiment we determined the concentration of cytokines CXCL12 and CCL2 in bone marrow (BM) and activated microglia in the brain. Nest complexity scoring, automated home-cage behaviour analysis (HCS) and Elevated Plus Maze testing (EPM) were used to detect any behavioural alterations. Recordings were made before and after a 15-minute period of R&L in groups of mice handled via TH, NAH-T or NAH-M.

## Results

BAT temperature significantly decreased in all handling groups following R&L regardless of handling method. There was a difference, at the limit of significance (p = 0.06), in CXCL12 BM content among groups. CXCL12 content in BM of NAH-T animals was similar to that found in Sentinels, the less stressed group of animals. After R&L, mice undergoing NAH-T and NAH-M showed improved body-weight maintenance compared to those exposed to TH. Mice handled via NAH-M spent a significantly longer time on the open arms of the EPM. The HCS results showed that in all mice, regardless of handling method, R&L resulted in a significant reduction in walking and rearing, but not in total distance travelled. All mice also groomed more. No difference among the groups was found in Nest Score, in CCL2 BM content or in brain activated microglia.

## Conclusions

Stress induced by a common restraint procedure caused metabolic and behavioural changes that might increase the risk of unexpected bias. In particular, the significant decrease in BAT temperature could affect the important metabolic pathways controlled by this tissue. R&L lowered the normal frequency of walking and rearing, increased grooming and probably carried a risk of low-grade neuro-inflammation. Some of the observed alterations can be mitigated by Non-aversive handlings.

## Introduction

Animal models used to investigate conditions like heart failure, stroke, hypertension or cancer, and to develop rehabilitation procedures, must be refined such that they deliver results with the maximum possible translational relevance. Only then can they be used to effectively develop therapeutic and rehabilitative strategies for humans suffering similar conditions. This means limiting undesirable effects that might generate bias. It has been shown that chronic psychological stress in laboratory animals can lead to CNS neuroinflammation [1–3], an effect that may contribute to experimental bias. Such biases or sources of experimental 'noise' are undoubtedly contributors to what has been described as a current reproducibility crisis in preclinical research [4]. The extent to which routine but nevertheless potentially highly stressful procedures might negatively impact on the quality of the results of studies has not been extensively investigated. Studies concerning the impact of restraint are typically chronic in nature and the period of capture can last at least one hour [5–7], and are meant to model methods which may induce behaviours that have relevance to the clinical conditions of post-traumatic stress and major depressive disorders [8, 9]. However, mice are more commonly restrained for a period lasting only several minutes for procedures such as blood sampling or to receive

injections, and during this time they are usually also exposed to abnormally high ambient lighting. Although it is likely that such shorter more acute restraint exposures are also highly stressful, only a few attempts have been made to address whether this represents a major problem [10–12]. Following the discovery that chronic psychological stress leads to neuroinflammation at a central level [1] our group wondered whether the same potentially confounding effect might arise following putatively less impactful procedures, such as acute restraint under bright light (R&L); a scenario mice often encounter in pharmacological studies, for example for repeated injections or blood withdrawals. If this were the case, then neuroinflammation, even low-grade, could lower the reliability of results, and ultimately lead to greater numbers of animals being needed [13]. This problem, addressed in the case of chronic psychological stress [1, 3, 14, 15], has not been sufficiently studied in the case of minor operative procedures. Therefore, in this study we investigated whether 15 minutes R&L affected CXCL12 and CCL2 content in the BM and activated microglia in the brain, and whether it also affected physiological homeostasis, like changes in BAT temperature, as detected with IRT, and body weight alteration. We also investigated possible effects on behavioural parameters.

A period of 15 minutes, incorporating bright lighting, was considered appropriate to model the above mentioned scenario (repeated injections or blood withdrawals); moreover, 15 minutes of restraint caused an increase in plasma corticosterone and ACTH levels, together with a significant rise of the anxiety levels, in rats [12], thus suggesting that also in mice 15 minutes of containment might affect physiological homeostasis.

CXCL12 and CCL2 were measured for their key role in the interplay between BM, microglia and immune mediators seen in chronic psychological stress neuroinflammation, where they are involved in the mobilization of inflammatory cells from the BM, followed by subsequent brain infiltration through the blood-brain barrier [1, 14–19] and undesirable central inflammation.

Regarding IRT, its benefit consists in the fact that it can be used unobtrusively in freely moving mice, thus representing a refinement compared to methods typically requiring placement of a transponder either onto or inside the body [20–24]. In our recent study using a model of spinal cord injury [21] we showed IRT to be effective for detecting links between post-surgical pain and increases temperature in the mice BAT, suggesting that surgical pain might influence BAT homeostasis and possibly systemic metabolism. We therefore wanted to assess whether also acute R&L stress might affect BAT temperature.

Also, the way mice are handled can make an important contribution to stress susceptibility, and handling them by the tail, which is still the most commonly used method [25], is now known to be aversive and unnecessarily stressful [26]. Therefore, a further aim was to verify whether, compared to standard TH, the NAH-T and NAH-M techniques might prevent or standardise acute R&L-induced stress.

NAH-T involves guiding mice into a Plexiglas tube, and once inside, transferring them between cages or other apparatus as necessary [26], while NAH-M consists in low-force, low-velocity stroking of the fur with brush resulting in MRGPRB4 G-protein-coupled receptor activation which has demonstrated anxiolytic effects [27, 28] and is found in social interactions involving grooming behaviors in mice [29].

Thereby, the hypothesis of the present study was that 15 minutes of R&L stress would produce neuroimmune and physiological alterations consistent with stress, and that the effects of this could be mitigated, and therefore welfare refined, via the use of two forms of non-aversive handlings. In particular, the primary endpoint was the CXCL12 chemokine content in the bone-marrow, [the decrease of this chemokine in BM indicates egression of potentially inflammatory cells from BM [1, 15]], and one of the main secondary endpoints were eventual changes in brown adipose tissue temperature detected by Infrared thermography. The

behavioural data were collected to have a complete overview within this exploratory study, despite being not the primary endpoints.

## Materials and methods

### Ethical statement

All procedures were approved by the Italian Institute of Health (Permit Number 55/2019-PR) according to 26/2014 Italian Law on the protection of animals used for scientific purposes. The manuscript was prepared according to the ARRIVE guidelines [30, 31].

### Husbandry

Forty-two male 8-week-old CD1 mice were supplied by Envigo srl (Milan, Italy). Only males were used to avoid the biological variable of sex. Animals were individually housed in polycarbonate cages (268x215x141 mm) with wood-shaving bedding (ENVIGO RMS) with free access to tap water and rodent feed (Teklad Global Diet 18%—ENVIGO RMS). Upon arrival they were allowed six days to acclimate to their new surroundings. Cages were open to the room environment and, when necessary for normal husbandry activities like cage cleaning, animals were handled two times a week by the staff involved in the experimental protocol. Strict adherence was made to the method of handling each mouse by its pre-assigned method, which was lifting by the tail in the TH and NAH-M groups and lifting by tunnel in NAH-T group. Room temperature was maintained at 20˚±2˚C with 55% ± 10% relative humidity and ventilated at 15–18 filtered air changes per hour. Animals were kept on a 12:12 light: dark cycle (lights on at 6am). Shredded paper strips were use as environmental enrichment [32]. Blood screening was performed on sentinel mice (not those involved in the experiment) every six months during the year, to certify the absence of endoparasites and ectoparasites in the animal facility, which is classified as conventional.

### Handling treatments method and experimental groups

1. NAH-M, n = 10. A cosmetics brush was used to apply NAH-M stimulation [miniature paintbrush Cadrim n˚ 21 [27]], whose softness was tested on the base of our clinical experience to match the force parameters indicated by Delfini and colleagues [33]. Stimulation was performed by gently stroking the back of the animal with the brush for 5 minutes, in a cephalocaudal direction, without removing the mouse from the cage. The application force and brush velocity fell within the range described by Vrontou et al [27]; approximately 3 cm/s and a force of 2.5mN. When necessary (e.g. for cage cleaning), animals were moved by picking them up by holding the tail.

2. TH, n = 10. As described by Gouveia and Hurst [26], animals were captured by the base of the tail and lifted and then supported on the experimenter's sleeve for 30 seconds (secs). The mouse was then released back into its home cage. The handler then stood back from the cage for 60 sec, and then repeated handling for a further 30 secs.

3. NAH-T, n = 10. Animals were lifted using a handling tunnel as demonstrated in an online video tutorial (https://www.nc3rs.org.uk/how-to-pick-up-a-mouse). They received a session of NAH-T handling as also described by Gouveia and Hurst [26]; i.e. once lifted using the handling tunnel (50 mm diameter, 100 mm long), mice were held for 30 secs, then placed back into their cage. The experimenter's hands were loosely cupped around the end of the tunnel to prevent the mouse exiting. The experimenter then moved away from the cage for 60 secs and then handled the mice a second time for 30 secs.

A group of twelve sentinel mice were provided with shredded paper strips for nesting but were not handled and did not undergo any procedures during the entire duration of the study. When necessary for cage cleaning, they were transferred by tail handling. These animals were used as controls for biochemical parameters and represent the less stressful condition of housing.

## Experimental design

The study design and order of data collection is depicted in Fig 1. The procedures were conducted during the light-on cycle (approximately between 7am and 3pm) in order to assess the impact of the R&L procedure as it is normally applied–i.e. during the day. In particular, in the vivarium light was on at 6 am and off at 6pm and the time when the behavioral tests and samples were collected was consistently during the light phase, at least one hour away from the transition phase. When necessary, caretakers entered the room at the same hour which was around 9 am. On arrival mice were randomly assigned using a fair coin toss to one of three handling groups, TH, NAH-T or NAH-M (n = 10 per group), or to Sentinel group (n = 12). Sentinels were provided with shredded paper strips for nesting, and apart from cage cleaning they were not handled and did not undergo any other procedures before sacrifice. They were exposed to the same environment the others mice were exposed to and were individually housed specifically on the same rack as the experimental animals. During cage cleaning they were transferred to their new cage via tail handling. They provided controls for the biochemical/neuroinflammatory parameters but no behaviour data were collected.

Once the acclimation period was complete, 3 to 4 mice from each handling group were tested each week, for a total of three study weeks. Mice in the TH and NAH-T groups were always handled according to their assigned method during transfers between the various types of test apparatus (details of this below). Mice in the NAH-M group were lifted by the tail for

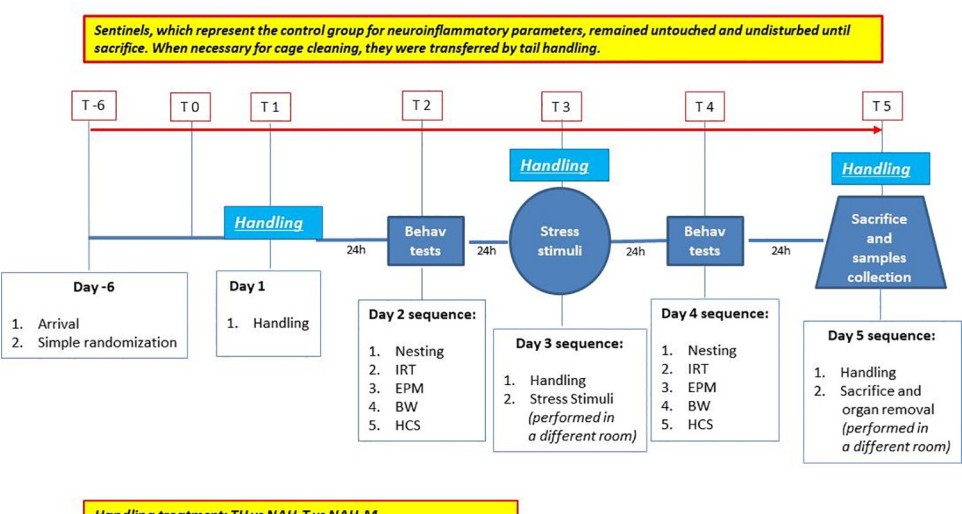

**Fig 1. Study design.** One complete test lasted 5 consecutive days. In the first two days, the different handling methods (TH, NAH-M and NAH-T) and basal behavioural tests were performed, followed by R&L stress stimuli during the 3rd day. The hour before stress stimuli, animals received a session of handling. The 4th day, 24 hours after stress, post-stress behavioural tests were performed and the last day, after another handling session, mice were sacrificed and organ collected. The entire study lasted 3 weeks, repeating the complete test 3 times.

this purpose. The three experimental groups underwent behavioural testing and application of R&L according to the following timeline:

- Day 1: One exposure to handling using TH, NAH-T or NAH-M

- Day 2: Baseline assessment of behaviour and anxiety status (between 7am to 3pm, according to the following test sequence;

  1. Nesting scoring (without disturbing the animals or touching the nest)

  2. Recording of BAT temperature via IRT

  3. EPM test (10 minutes)

  4. Body weight recording

  5. HCS Behaviour recording (in a separate room)

- Day 3: Handling repeated before placing each mouse into the R&L device for 15 minutes (described below, R&L procedure, performed in a separate room).

- Day 4: Behavioral testing repeated as on Day 2 (with the timing of events as far as possible matching Baseline timing);

- Day 5: Sacrifice day, with the following sequence of actions between 7am and 3pm:

  1. Handling

  2. Sacrifice and organ removal (in a separate dedicated surgical room)

Tissues were harvested for later processing as described below. Fourteen animals were examined per week, for a total of three weeks of experiments.

## Experimental procedures

**Stress test.** R&L Stress consisted of one 15-minute period of restraint in a small Plexiglas tube (Mouse Tailveiner Standard, diam. 25 mm, 2 Biological Instruments–Varese, Italy). During this time they had a 400–500 lux light source directed at them from their left at a distance of 20 cm [Bouwknecht [34]; Gameiro [12]].

**Nesting building.** Nest quality was recorded by taking ~30 secs of video footage and then 10 photographs using a digital camera (Sony Cybershot W830). After opening each cage and without disturbing the mouse, approximately 30 seconds of video footage of the completed nest was taken (without continuously following the construction process). The statistical analysis was therefore conducted on nest score at 24 h. Ten photographs were also taken, according to the method of Hess [32]. Mice were considered to have built a nest only if the nesting material contained a hole in the middle and there was a clear sign of a depression in the centre. Mice that had not done so received a score of 0. If there was an interaction with the nesting material, consisting in shredded paper strips, the score was 1 (Fig 2A). When the nest was flat, with a cavity in the middle but with no, or incomplete, walls, nest score was 2 (Fig 2B). Score 3 corresponded to a nest with walls forming a "cup" shape (Fig 2C) and a score of 4 was awarded if the walls reached the widest point of an imaginary half of a sphere (Fig 2D). Finally, a nest forming a complete dome with walls completely enclosing the nest was given a score of 5 (Fig 2E).

**IRT.** A microbolometric high resolution infrared camera (FLIR A65 model; 640 X 512 pixel sensor) was placed on a tripod 60 cm above the open cage. Mice were continuously recorded while freely moving for 3 minutes. Three still images were chosen from the IRT

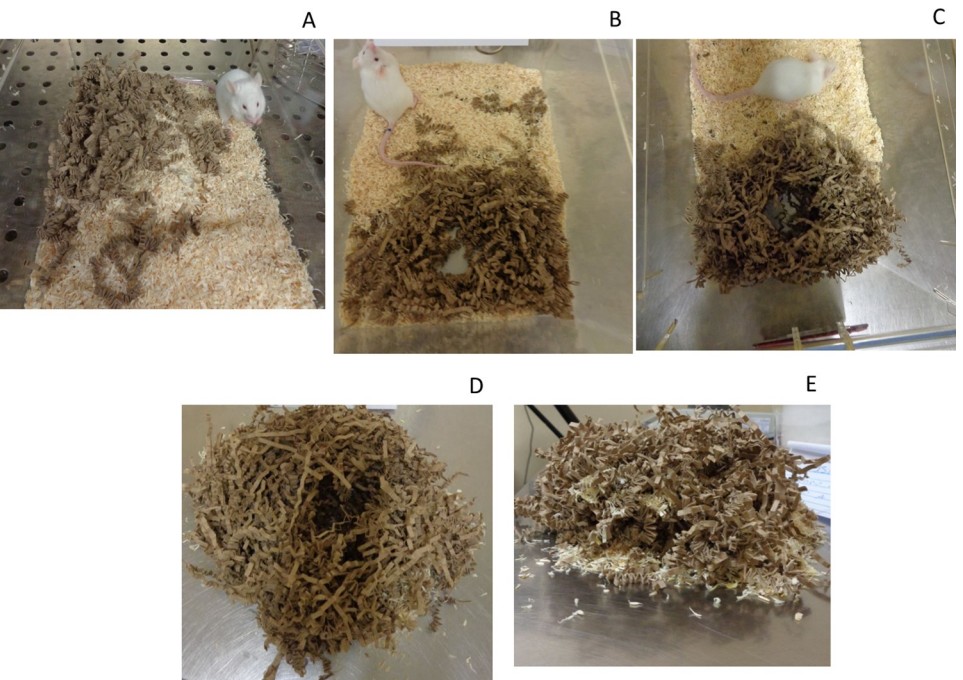

**Fig 2. Nesting.** Example of nest building. Nest score from 1 (A) to 2, 3, 4 and 5 (B to-E, see text for detailed description). There was no difference in nest score between the groups, either before stress, or after stress test.

video footage for each mouse; selected when the image was in focus and the BAT tissue area, chosen on the base of previous studies [21, 35, 36], was clearly visible. The median value of the maximum temperature in this area was calculated using two software: FLIR Tools (FLIR Systems, https://www.flir.it/) and IRT Analyzer (GRAYESS software, https://www.grayess.com/).

**EPM.** The EPM is a well-established method for anxiety assessment [37]. The EPM platform was 50 cm from the ground and consisted 2 two open and 2 close arms facing one another. Each arm was 50 cm length and 10 cm wide. The closed arm walls were 30 cm high. After lifting each mouse by the appropriate handling method, they were placed into the centre of the maze facing an open arm. They were then recorded using a Sony Digital Handycam video camera placed on a tripod 50 cm above the apparatus for 5 minutes.

**HCS.** Automated behaviour analysis software (www.cleversysinc.com) was used to record the spontaneous unconstrained behaviour of the mice. The setup for this has previously been described [38]. Briefly, mice were filmed for 10 minutes using a Sony Digital Handycam video camera placed on a tripod approximately 30 cm from the cage front. The cages were lit from behind using a neon light covered with a thin sheet of paper, to evenly distribute the light, thus creating the contrast necessary for subsequent analysis with the HCS software [38]. Recordings lasted ten minutes. The total time needed to undertake each of the 3 types of recording (IRT, EPM and HCS) was approximately 23 minutes; including roughly 10 minutes for handling and moving mice from one apparatus to the next.

**Anaesthesia and sacrifice.** Each mouse was anaesthetized by intraperitoneal injection of 40 mg/kg Tiletamine/zolazepam (Zoletil 100) and 8 mg/kg xylazine (Xilor), as recommended by 26/2014 Italian Law on the protection of animals used for scientific purposes. Once fully unconscious (verified by toe-pinch method) they were killed by surgical cervical dislocation (surgical cutting of the cervical vertebrae), to avoid brain damage. Femur bones and brain were collected.

**Cytokine quantification in BM.** Isolation of femur bone from mice was adapted from the method of Madaan et al [39]. In brief, isolated femur was rinsed in cold PBS, both ends were cut and BM was flushed by 29G × ½ needle. BM was flushed from the other end by inverting the femur, directly in the Eppendorf tube containing RIPA extraction buffer and immediately stored in dry ice. ELISA assays for CXCL12 and CCL2 cytokine quantification were performed by LABOSPACE Srl (Via Ranzato, 12–20128 –Milan, Italy), by using R&D MCX120 Mouse CXCL12/SDF-1 alpha Quantikine ELISA kit and R&D MJE00B Kit Mouse CCL2/JE/MCP-1 Quantikine ELISA Kit, following kit instructions. In particular, for Mouse CXCL12/SDF-1 alpha Quantikine ELISA Kit, the Assay Range was defined from 0.2 to 10 ng/mL in tissue lysed and relative sensitivity 0.069 ng/mL. Vendor evaluated Intra–and Inter assay precision at 3.7–5.1% and 7.2–7.5%. Vendor indicated a recovery of above 100% in the dilution's ranges indicated for assay.

For Mouse CCL2/JE/MCP-1 Quantikine ELISA Kit, the assay range was defined from 7.8 to 500 pg/mL in tissue lysed with sensitivity 0.666 pg/mL. Vendor evaluated Intra–and Inter assay precision at 2.5–4% and 5.1–7.5% which indicated a recovery of above 100% in the dilution's ranges indicated for assay.

Both ELISA assays are specific for mouse species with < 0.5% cross-reactivity observed with available related molecules. Each ELISA assay included one blank and 7 points for the standard curve. Each plate was loaded with 40 samples. All samples were run in duplicate. Data were validated by testing laboratory and QC was performed. A Four Parameter Curve was applied for standard samples fitting. Values of the standard curve were compared with the values provided by the ELISA vendor not exceeding a $R^2$ of 10% and all of the above parameters were applied on at least 90% of the standard curve values. All CVs of unknown samples met specifications of both vendor's ELISA.

**Activated brain microglia assessment.** Activated microglia in the brain were analysed using immunohistochemistry according to manufacturer instructions. After buffered formalin fixation, coronal cutting of the whole brain was made and paraffin embedding was performed. Three micrometer (3 μm) thin sections were cut and stained with Haematoxylin-Eosin and Iba-1 antibody (Abcam Microglia marker panel Ab226482). The section was pre-treated using heat mediated antigen retrieval with Ethylenediaminetetraacetic acid (EDTA) for 3 cycles of 5 minutes each in microwave oven. The section was then incubated with recombinant Iba-1 antibody, 1/2000 dilution, for 2 hours at room temperature. Abcam Iba-1 is a recombinant, non-pre-absorbed antibody. In order to obtain a better result, we used the MACH4 HRP polymer, a very effective amplification system, as a secondary antibody. DAB was used as the chromogen. Negative control was performed by removing the recombinant Iba-1 antibody. The section was then counterstained with haematoxylin and mounted.

Two areas were evaluated for assessment of the activated brain microglia: Paraventricular nucleus (PVN) [1] and the entire first section posterior to the midpoint of the brain. In order to define the brain areas, we used the atlas of Paxinos and Franklin [40]: the entire brain section posterior to the midpoint was taken 2 mm posterior to the bregma. According to the Atlas, we identify PVN just beneath the 3rd ventricle, at the following coordinates: -1.5 rostrocaudal; +/- 0.1 lateral (i.e. millimeters from bregma and from the midline, respectively). The histological slides were digitalized by Hamamatsu Nanozoomer scanner; then Iba-1 positive cells were counted using the ImageJ software as follows: image was transformed into black and white and was subjected to threshold filtering shape and size of the staining in order to identify cell nuclei; in order to measure the Iba-1 positive cells in a non-biased manner, areas and signals were automatically counted by the software and researchers performing histology were blinded to the treatment.

The two areas of interest, the PVN or the whole brain section, were measured in mm$^2$, then Iba-1$^+$ cells density was calculated for each sample. Due to the small size of the PVN, Iba-1 positive cells in this region was expressed differently from those in the entire brain sections. For the PVN, the calculation was the number of Iba-1$^+$ cells/0,08 mm$^2$, whereas in the entire brain section it was the number of Iba-1$^+$ cells/mm$^2$. The entire brain section results were then multiplied for the same factor (x100), to avoid decimals numbers lower than 1.

## Statistical analysis

The numerosity within the groups derived from a power calculation for the primary endpoint which was CXCL12 chemokine concentration in the BM (software available at http://www.biomath.info/). On the base of the literature [1, 41], a standard deviation of 50 and a difference of 70 pg/ml was expected between groups to have an alpha value less than 0,05 and a power of 0,80.

The description of the groups of interest with respect to clinical characteristics has been performed by means of median and interquartile range for continuous covariates and by means of absolute and relative frequencies for categorical ones.

Due to the exploratory aim of this study and the small sample size a non-parametric approach was used to investigate the differences within and between groups and we did not perform significance testing to avoid inflation of type 1 error, so generating an excess of false positive signals. To evaluate differences between two groups and intra group (before vs after stress) Wilcoxon-Mann-Whitney test and Wilcoxon signed rank sum test have been performed respectively. Moreover, Kruskal-Wallis (KW) test was used to study the differences among groups. Furthermore, eta-squared based on the H-statistic was calculated to evaluate the percentage of variance in the dependent variable explained by handling groups [42]. This index ranges from 0% to 100% and values between 0% to 6% indicate small effect, from 6% to 14% moderate effect and more than 14% large effect [43]. Statistical analysis was generated using SAS/STAT software, version 9.4 of the SAS system for Windows. Range plots with capped spikes were generated using Stata Software, version 15.1 (StataCorp. 2017. College Station, TX: StataCorp LLC). All statistical tests are interpreted at the 5% significance level considering two-sided test.

## Results

### IRT

There was no significant difference in temperatures among the handling treatment groups, but the temperature significantly decreased in all groups after stress test in BAT area. In particular, temperature of NAH-M treated mice before stress was 30,5˚C (range IQ: 30–30,7) and after stress was 29,5˚C (range IQ: 29–30), p = 0,037. In the NAH-T handled animals temperature decreased from 30,4˚C before stress test (range IQ: 30,26–30,66) to 29,2 after test (range IQ: 28,5–29,7), p = 0,0097. Control animals, TH, decreased their temperature from 30,2˚C to 29,4˚C (range IQ: 29,8–30,7 before stress and range IQ: 29,2–30 after stress), p = 0,019 (Fig 3).

### BM CCL2 and CXCL12 content

There was no significant difference in CCL2 content in BM among groups [TH 8,014 pg/ml (range IQ:4,352–9,568), Sentinel mice 9,858 pg/ml (range IQ:6,055–10,663), NAH-T 6,411 pg/ml (range IQ:3,875–11,056) and NAH-M 7,597 pg/ml (range IQ:3,935–11,689) H of KW = 0.71, p-value KW = 0,87, Eta$^2$ = 0%] (Fig 4A).

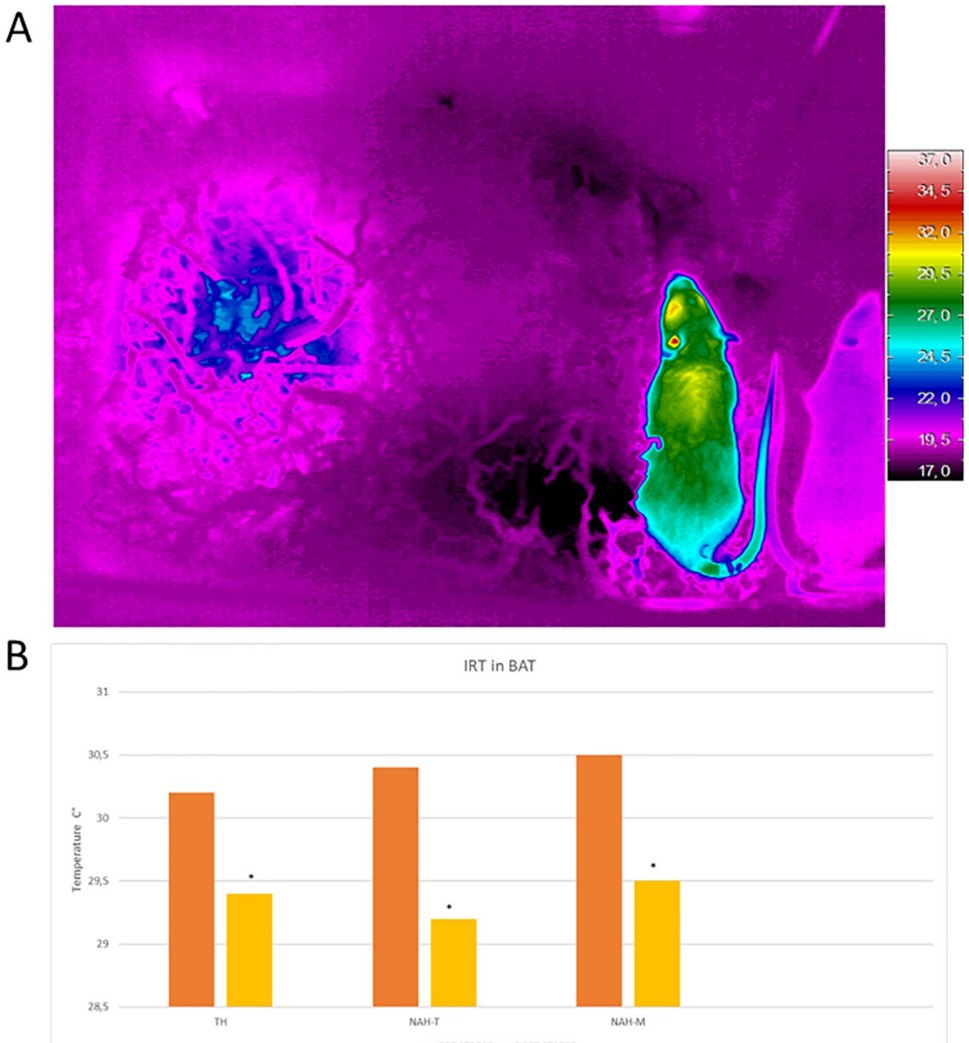

**Fig 3. IRT image and measurements.** A-Example of IRT image of a mouse: the part in yellow on the dorsal region is the BAT area. B-IRT measurements: after R&L stress test, the temperatures recorded in the BAT area in all experimental groups were lower than those recorded at baseline, suggesting that this stress stimuli might affect BAT homeostasis, with possible consequences on metabolism (*p< 0,03 vs baseline).

There was a difference, at the limit of significance (p = 0,06), in CXCL12 BM content among groups. Interestingly, CXCL12 content in BM of NAH-T animals was similar to that found in Sentinels, the less stressed group of animals [TH 1,528 ng/ml (range IQ:1,344–2,025), Sentinel mice 2,531 ng/ml (range IQ:1,631–3,122), NAH-T 2,346 ng/ml (range IQ:1,762–2,741), and NAH-M 1,114 ng/ml, (range IQ:1,036–1,546), H = 7.31, p- value KW = 0,06, Eta$^2$ = 13%] (Fig 4B).

## Immunohistochemistry

Considering observation by Ataka et al [1], who found stress-related activated microglia in PVN, this was analysed first, then the entire brain slice was analysed. Histological analysis revealed normal brain architecture and activated microglia were present, but there were no significant group differences.

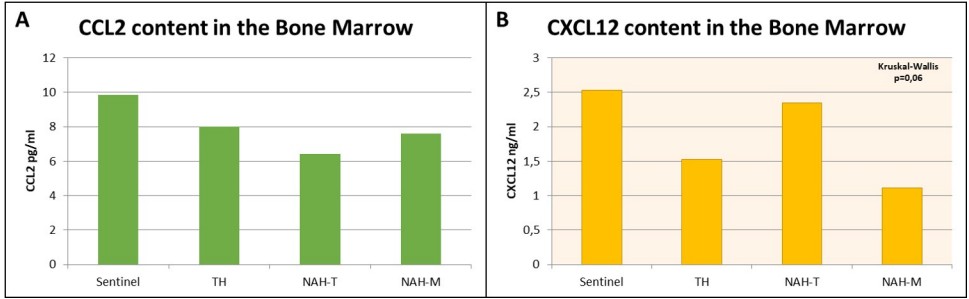

**Fig 4. Cytokines content in the bone marrow.** A) CCL2 content: no significant difference was found among groups. B) R&L stress did not cause massive neuroinflammatory activation, despite a trend of CXCL12 decrease in TH and NAH-M groups with respect to Sentinels (Kruskal-Wallis test: p = 0,06), which might indicate an inflammatory process at its final stage.

[*PVN*: TH 5,29 Iba1$^+$cells/0,08mm$^2$ (median area) (range IQ:2,84–7,09), Sentinel 3,47 Iba1$^+$cells/0,08mm$^2$ (range IQ:2,17–6,39), NAH-T 5,39 Iba1$^+$cells/0,08mm$^2$ (range IQ:3,65–5,71) and NAH-M 4,15 Iba1$^+$cells/0,08mm$^2$ (range IQ:2,88–5,08) H of KW = 2.51, p-value KW = 0.473, Eta$^2$ = 0%].

[*Entire Brain Slice*: TH 1,936x10$^{-2}$ Iba1$^+$cells/mm$^2$ (range IQ:1,145–3,051) Sentinel 1,4x10$^{-2}$ Iba1$^+$cells/mm$^2$ (range IQ:0,933–2,584) NAH-T 1,420x10$^{-2}$ Iba1$^+$cells/mm$^2$ (range IQ:0,953–2,891) NAH-M 1,373x10$^{-2}$ Iba1$^+$cells/mm$^2$ (range IQ:0,771–2,583) H of KW = 1.69, p-value KW = 0.638, Eta$^2$ = 0%] (Fig 5 and Table 1).

## Body weight

After stress, there was a difference in percentage of body weight gain among groups (H of KW = 6.886 p = 0.032, Eta$^2$ 18%). In particular, NAH-T and NAH-M were associated with an increase in body weight with respect to TH, suggesting that Non-aversive handlings might protect mice from the lack of weight gain seen in TH group. [NAH-M was 37 gr before stress (range IQ:36–40) and 38 gr after stress (range IQ: 36–40), TH was 36 gr (range IQ:35–37,5) before stress and 36 gr (range IQ: 35,5–37,5) after stress, NAH-T was 38 gr (range IQ: 37,25–40) before stress and 39 gr (range IQ: 38–40,75, after stress] (Fig 6 and Table 2).

## Behavioural tests

**EPM.** After R&L stress, there was a significant increase in the ratio post/pre of time spent on open arm in mice treated with NAH-M with respect to NAH-T and TH handled groups (H of KW = 6.4585, p-value KW = 0.0396, Eta$^2$ = 18%) (Fig 7 and Table 3). No difference was found before and after stress in TH and NAH-T groups [TH = 40 secs (range IQ:31–75) before stress and 54 secs (range IQ:40–70) after stress, p = 0,89; NAH-T = 64 secs (range IQ:50–79) before stress and 62 secs (range IQ:33–85) after stress; p = 0,49]. No difference was found before and after stress neither in the number of open arm entries nor in the number of protected stretch attends in any of the experimental groups (Table 3).

**HCS.** R&L stress caused a significant reduction in walking and rearing frequency (p = 0.04, p = 0.027, respectively), an increase in grooming duration (in average bout duration seconds; p = 0.0004) and frequency (p<0.0001) in all animals (Fig 8 and Table 4). Handling method had no significant effects on these changes. No difference was seen in total distance travelled.

**Nest score.** There was no difference in nest score between the groups, either before stress, or after stress test (Table 5). Before stress test, 24 hours after nesting material availability TH

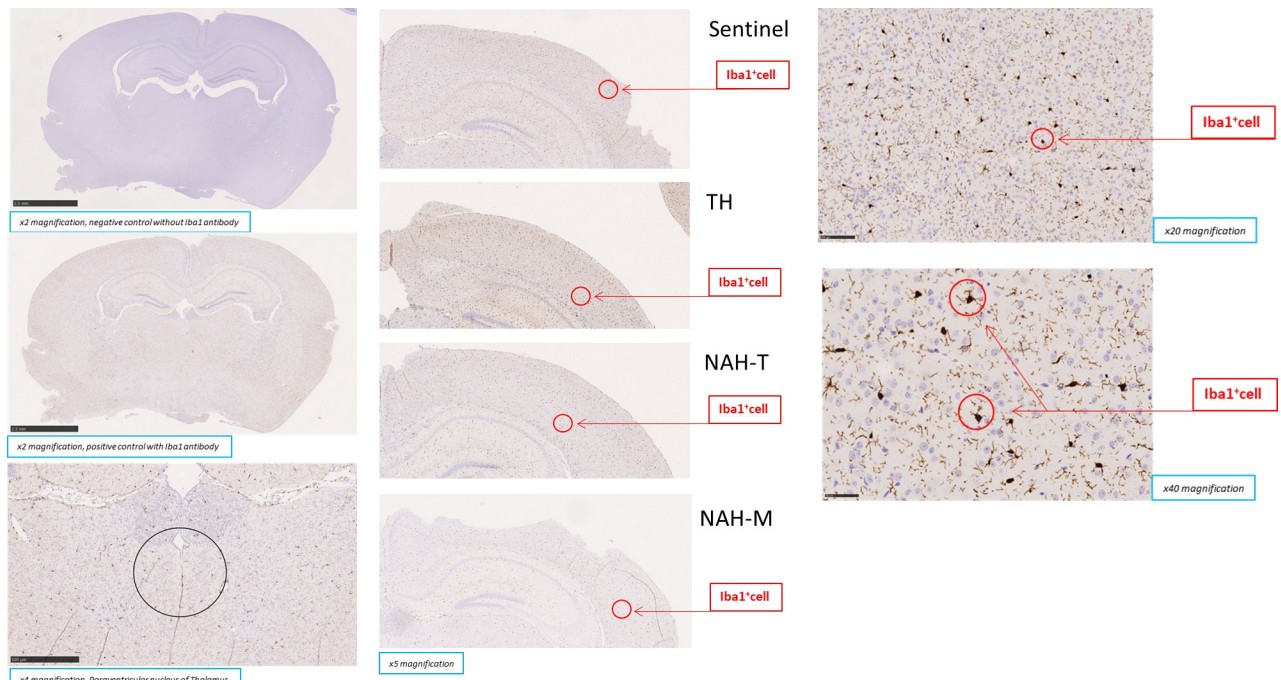

**Fig 5. Activated microglia in the brain.** Histological analysis of the brain. Presence of activated microglia was found (Iba1+ cells in the red circle), but there was not significant difference between R&L stressed mice, regardless of handling, and Sentinels, suggesting that if an inflammatory process occurred in BM, it did not reach the central nervous system. Left: above a x2 magnification of negative control without Iba1 antibody, in the middle a x2 magnification of positive control with Iba1 antibody, below a 4x magnification of PVN (area inside the circle, see methods for stereotaxic coordinates). Center: a representative photomicrograph for each handling group (x5 magnification). Right: a series of images with different degrees of magnification of Iba1+ cells (x20 and x40).

mice displayed a score of 4,75 (range IQ 4,75–5), NAH-T a score of 4,5 (range IQ 0–5), and NAH-M a score of 4,75 (range IQ 4,5–5). H of KW = 3.00, p-value KW = 0.223, Eta$^2$ = 4%. After stress test, 24 hours after nesting material availability TH mice displayed a score of 5 (range IQ 4,75–5), NAH-T a score of 4,75 (range IQ 4,75–5) and NAH-M a score of 5 (range IQ 4,75–5). H of KW = 1.93 p-value KW = 0.381, Eta$^2$ = 0%.

**Table 1. Immunohistochemistry staining.** Median values of Iba1+ cells number, with interquartile ranges (25° ptcl-75° ptcl), present in the PVN (H of KW = 2.51, p-value KW = 0.473, Eta$^2$ 0%) and in the entire brain slice (H of KW = 1.69, p-value KW = 0.638, Eta$^2$ = 0%) after immunohistochemistry staining.

| **PVN** | | | | |
|---|---|---|---|---|
| Treatment | **n** | **Median Iba1+cells/0,08mm$^2$** | **25° pctl** | **75° pctl** |
| Sentinel | 9 | 3,47 | 2,17 | 6,39 |
| TH | 10 | 5,29 | 2,84 | 7,09 |
| NAH-T | 10 | 5,39 | 3,65 | 5,71 |
| NAH-M | 10 | 4,15 | 2,88 | 5,08 |
| **Entire Brain** | | | | |
| Treatment | **n** | **Median Iba1+cells/mm$^2$** | **25° pctl** | **75° pctl** |
| Sentinel | 9 | $1,4 \times 10^{-2}$ | 0,933 | 2,584 |
| TH | 10 | $1,936 \times 10^{-2}$ | 1,145 | 3,051 |
| NAH-T | 10 | $1,420 \times 10^{-2}$ | 0,953 | 2,891 |
| NAH-M | 10 | $1,373 \times 10^{-2}$ | 0,771 | 2,583 |

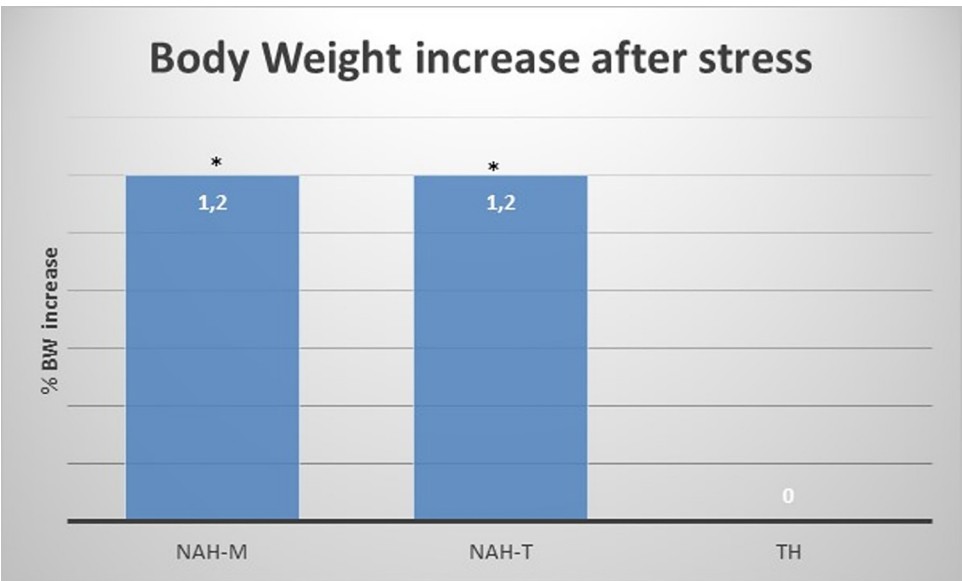

**Fig 6. Body weight.** Difference in percentage of body weight gain among groups. NAH-T and NAH-M were associated with an increase in body weight with respect to TH, suggesting that Non-aversive handlings might protect mice from the lack of weight gain seen in TH group (H of KW = 6.886 p = 0.032, Eta$^2$ 18%).

## Discussion

Restraining mice is a common requirement in biomedical research. This is likely to be highly stressful and to influence research findings, and the number of studies assessing if this is a problem is steadily rising [26, 44–46]. However, the focus of these has been on stress-indicative behavioural alterations following restraint by manual handling (e.g., scruffing), and whether these can be minimised using non-aversive (tunnel) handling. There have been no studies to determine the effects of another restraint method that is common for intravenous substance injection, i.e., restraint in an injection tube. The aim was to determine whether this not only has harmful behavioural, but also physiological and neurological consequences. As a secondary aim we also wished to determine if the effects of this type of restraint can be minimised using tunnel handling, but we also included a less well known potentially non-aversive option: mechanoceptive handling.

The choice to work only with male mice was based on evidence that sex can influence responses to handling [45, 47]. In their original article Hurst and West [44] found that although both sexes had lowered anxiety following tunnel handling, females were more responsive. It was later confirmed that male mice are more sensitive to handling-related stress [45, 47]. As our study was exploratory in nature we wished to avoid this potential sex-related influence.

**Table 2. Percentage of body weight increase after R&L stress, (H value of Kruskal Wallis = 6,88; p = 0,032, Eta$^2$ 18%).** NAH-T and NAH-M were associated with a significant increase in BW after stress with respect to TH. TH handled animals showed no body weight increase after R&L stress, therefore the BW percentage value was 0.

| BW percentage ((Post -pre)/pre)*100 | H of KW = 6.88 p-value = 0.03, Eta$^2$ = 18% | | |
|---|---|---|---|
| *TREATMENT* | *Median* | *25° pctl* | *75° pctl* |
| **NAH-M** | 1.21951 | 0 | 2.7027 |
| **TH** | 0 | 0 | 0 |
| **NAH-T** | 1.21951 | 0 | 2.63158 |

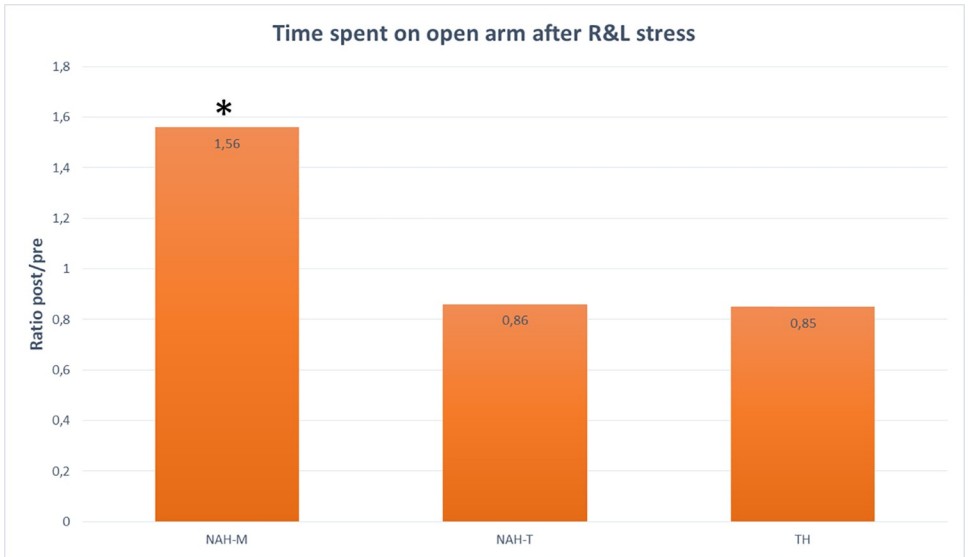

**Fig 7. EPM test.** Time spent on open arm after R&L stress expressed as ratio of post/pre-stress. Mice treated with NAH-M spent a significantly longer time on open arm after stress, (H value of Kruskal Wallis = 6,458, p = 0,0396, $Eta^2$ = 18%.) suggesting a beneficial effect on anxiety in this experimental group.

A significant and unexpected finding was that R&L stress caused a significant decrease in BAT temperature in all animals. Several studies have shown that decreased peripheral body temperature (usually caused by adrenergic mediated vasoconstriction) is a fear and stress-related phenomenon, especially in the case of long-lasting inescapable restraint [48–51]. In our

**Table 3. EPM test: A) time spent on open arm, number of open arm entries and protected stretch attends before and after R&L stress.** Median values with interquartile ranges (25° ptcl-75° ptcl.) are reported. Wilcoxon signed rank sum test: *p = 0.025 vs pre-stress value-. B) ratio post/pre increase of time spent on open arm, H value of Kruskal Wallis = 6,458, p = 0,0396, $Eta^2$ = 18%.

A

| EPM results | Handling | Pre-stress | p25 | p75 | Post-stress | p25 | p75 |
|---|---|---|---|---|---|---|---|
| Time spent on open arms (sec) | NAH-M | **47,5** | 36 | 67 | **90**[*] | 58 | 131 |
| Time spent on open arms (sec) | TH | **40** | 31 | 75 | **54** | 40 | 70 |
| Time spent on open arms (sec) | NAH-T | **64** | 50 | 79 | **62** | 33 | 85 |
| Number of open arm entries | NAH-M | **4** | 3 | 6 | **4** | 3 | 6 |
| Number of open arm entries | TH | **3,5** | 3 | 5 | **3** | 3 | 4 |
| Number of open arm entries | NAH-T | **5** | 4 | 6 | **4** | 2 | 6 |
| Number of protected stretch attend | NAH-M | **2,5** | 1 | 4 | **3** | 2 | 4 |
| Number of protected stretch attend | TH | **2,5** | 0 | 4 | **0** | 0 | 2 |
| Number of protected stretch attend | NAH-T | **1** | 1 | 2 | **1,5** | 0 | 4 |

B

| Ratio post/pre of time spent on open arm | | | |
|---|---|---|---|
| H of KW = 6.46, p-value KW = 0.04, Eta2 = 18% | | | |
| Handling | Median | 25° pctl | 75° pctl |
| NAH-M | **1.56137** | 0.8773 | 3.23529 |
| TH | **0.85281** | 0.58246 | 1.44921 |
| NAH-T | **0.86207** | 0.57971 | 1.09231 |

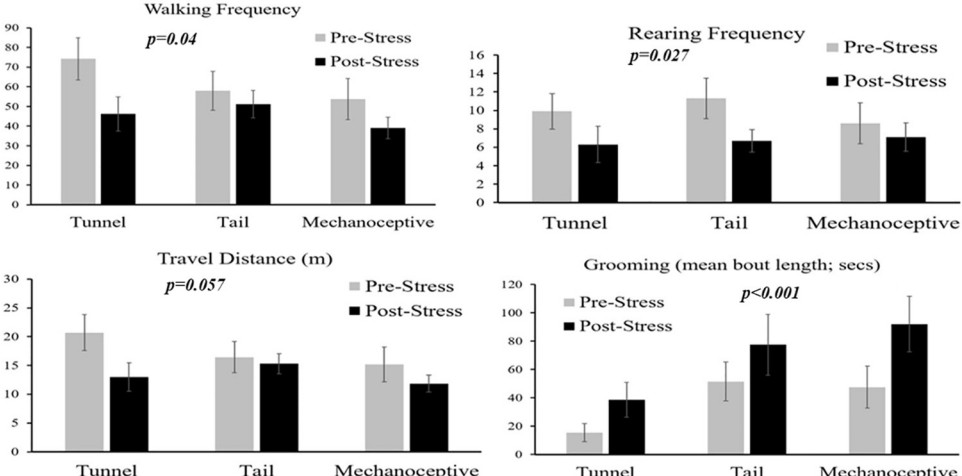

**Fig 8. HCS.** The results of automated behaviour analysis (with HCS) showing behaviour from before and after exposure to restraint stress in mice undergoing each method of handling (NAH-T, TH or NAH-M). P-values refer to comparison between pre-post R&L stress on all animals. Walking and Rearing frequencies significantly declined following stress, although the Total Distance mice moved (metres) tended to decline, not significantly. The average duration of bouts (in seconds) of Grooming increased. Handling method had no significant effects.

case R&L was applied for a moderately long period so it is reasonable to assume that the temperature reductions we recorded were directly attributable to the stress test. Several studies have shown that BAT has an important role in bone health and in glucose homeostasis [35, 36], consequently the exposure to animals of stimuli affecting this tissue might create bias in other types of physiological data. Thus, those studies using procedures similar to ours would be advised to consider the impact that this type of restraint may have when interpreting findings. That said, contrary to what has been observed in models eliciting chronic psychological stress [1], the present procedure had relatively little impact in terms of neuroinflammatory activation. However, there was a trend of inflammation in the BM which might reach significance by anticipating the measurement at a time closer to R&L stress. In fact, 48 hours after R&L two groups of animals (TH and NAH-M) presented decreased CXCL12 content in the BM with respect to Sentinels, a sign of possible progenitor cells egression and differentiation in inflammatory cells [15, 18]. The difference among groups was almost significant (p = 0,06), but suggested an inflammatory process at its final stage, or at least at a later stage. Only NAH-T handled mice showed a CXCL12 content in the BM like that seen in Sentinels; potentially underlying the beneficial effect of tunnel handling on welfare. Further investigations are needed to determine the timescale of this inflammation more precisely.

R&L caused anxiety-like behaviour and reduced body weight that were to a limited extent modulated by handling. According to our EPM results NAH-M seemed to reduce anxiety, and

**Table 4. HCS median values with interquartile ranges (25˚ ptcl-75˚ ptcl).** Irrespectively to handling method, R&L stress caused a significant reduction in walking and rearing frequency, and an increase in grooming duration and frequency in all animals.

| HCS VALUES | | PRE-STRESS | | | | POST-STRESS | | | |
|---|---|---|---|---|---|---|---|---|---|
| Variable | n | Median | 25˚ pctl | 75˚ pctl | n | Median | 25˚ pctl | 75˚ pctl | P-value |
| Allwalk | 30 | 71 | 50 | 87.5 | 30 | 47.75 | 31 | 58 | 0.0445 |
| Groom_frequency | 30 | 2 | 0 | 4 | 30 | 5.5 | 2 | 8 | <0.0001 |
| RearUp_frequency | 30 | 10 | 3 | 15 | 30 | 5 | 2 | 11 | 0.027 |
| TravDist_frequency | 30 | 19.62 | 13.2 | 24.79 | 30 | 14.52 | 11.29 | 17.13 | 0.057 |
| Groom_duration | 30 | 22.3 | 0 | 56.68 | 30 | 59.98 | 19.2 | 101.96 | 0.0004 |

**Table 5. Nest score increased in 24 hours in all experimental groups, both before and after R&L stress test, suggesting that 15 minutes of acute R&L did not affect nest building ability.** In particular, before stress test, 24 hours after nesting material availability, TH mice displayed a score of 4,75, NAH-T a score of 4,5 and NAH-M a score of 4,75 (H of KW = 3.00, p-value KW = 0.223, Eta$^2$ = 4%). After stress test, 24 hours after nesting material availability TH mice displayed a score of 5, NAH-T a score of 4,75 and NAH-M a score of 5 (H of KW = 1.93 p-value KW = 0.381, Eta$^2$ = 0%).

| NEST SCORE | | |
|---|---|---|
| PRE-STRESS: H of KW = 3.00, p-value KW = 0.223, Eta$^2$ = 4% | | |
| POST-STRESS: H of KW = 1.93 p-value KW = 0.381, Eta$^2$ = 0% | | |
| **Handling** | **PRE-STRESS** | **POST-STRESS** |
| **NAH-M** | 4.75 | 5 |
| **TH** | 4.75 | 5 |
| **NAH-T** | 4.5 | 4.75 |

compared to tail-handled mice, both NAH groups were protected from significant post-procedural weight loss. Mechanoceptive handling has a strong clinical background both in animal and human models [27, 52–55]. It relies on activation of the unmyelinated C-fibers polymodal afferent system. C-fibers differentiate in free tactile arborizations on the skin creating a secondary touch system that is specifically processed by the interoceptive system [56]. Due to the strong ties between human pain perception and anxiety, intriguingly, and perhaps unsurprisingly, both human and animal studies have shown that interoceptive touch (involved in NAH-M) can modulate (usually reduce) chronic stress and anxiety [27], pain severity [33, 57], and can improve heart rate variability [58]. It also has a prominent role in social and emotional bonding [59] and in modulating the endogenous u-opioid system [60]. Although our data did not provide such firm evidence of an improvement in the response of mice undergoing NAH-M, they were still consistent with these previous findings. As such, it seems that in some circumstances it could have potential to be an alternative to tunnel handling to at least partially protect animals from unnecessary stress.

Despite substantial evidence that NAH-T can be an effective means of anxiety control [45, 46], it is still used far less frequently than tail handling [25], probably because it is perceived as being more time consuming. While this aspect may be particularly important in the routine of cage cleaning, NAH-T use during experimental procedures could have considerable advantages. Handling mice using a tunnel or using cupped hands rapidly reduces stress susceptibility [26, 45], can mitigate against stress due to injections or anaesthesia and can minimise stress following identification procedures [61]. As far as reproducibility of findings is concerned, non-aversive handling may be an important means of improving the precision of research findings [46, 62, 63].

Being similar to the restraint methods often needed for substance administration, an important study objective was to try to determine the severity of stress or anxiety caused by R&L. We found physiological/metabolic and behavioural alterations consistent with stress, including reduced BAT temperature, reduced spontaneous movements (rearing, walking) and increased grooming. However, despite these changes it had virtually no impact on the ability to mice to nest-build; an outcome that is more commonly associated with more severely impactful disease states or procedures such as surgery [64]. The neuroinflammatory response was also less severe than in conditions of chronic psychological stress [1]. These preliminary data therefore indicate that R&L probably induced stress of moderate severity. Nevertheless, they also indicate that what may be perceived to be relatively ordinary procedures still carry a risk of experimental bias, in which case the use of non-aversive handling methods, especially NAH-T seems a sensible strategy to attempt to mitigate against that risk.

## Study limitations

In the present work it was not possible to measure peripheral corticosterone and plasma cytokines levels since the most common blood collection method for this, cardiac puncture, would likely have influenced our assessment of neuroinflammation; which was one of our principal measurements. Blood collection via the facial vein was an alternative possibility and has less impact of on cytokine up-regulation [65], but does not allow taking of the necessary blood volume for both corticosterone and plasma cytokines measurements.

Our assessment of behavioural differences may have been aided had we testing during the night when mice are more active. For example, it has been shown that superior EPM data may be obtained from animals kept on a reverse light cycle [37]. However, our study was meant as a 'real world' investigation of the impact of stress as it is normally applied (during the day) and to investigate the risk of increased bias potentially resulting from this, and if this risk can be ameliorated by lowering stress using non-aversive handling methods. Finally, previous EPM findings have indicated that there should be an approximate correlation between the number of open arm entries and time spent on the open arms. We cannot explain such low numbers of open arm entries but note that it is not unusual. For example, Henderson et al. [46] reported an average of less than 1 open arm entry following tail-handling, increasing to only 3.5 in mice handled non-aversively (using a tunnel). In their study the corresponding percentage of time mice in each handling group spent on the open arms was only 7 and 18%, respectively. In our study, ignoring the relatively minor effect that handling method had, the equivalent values were 5.6% before stress vs. 7.6 afterwards. We also checked to see if in our data there were parallel changes in open arm entries and open arm time. The proportion of entries vs. time was between 3–4 before handling and between 5 and 7 afterwards. We admit that these values are still low, but they do parallel one another. Nevertheless, the behavioural outcomes were subsidiary to our main objective of exploring a possible neuroinflammatory effect of acute restraint under bright light. Handling method, and whether it might ameliorate the response to this stressor was secondary.

## Acknowledgments

Authors are grateful to Prof. Grzegorz Bilo for his support.

## Author Contributions

**Conceptualization:** Veronica Redaelli, Fabio Luzi, Pietro Zerbi, Daniele Di Lernia, John Vincent Roughan, Luca Porcu, Davide Soranna, Laura Calvillo.

**Data curation:** Veronica Redaelli, Paolo Cappella, Pietro Zerbi, Daniele Di Lernia, John Vincent Roughan, Luca Porcu, Davide Soranna, Laura Calvillo.

**Formal analysis:** Veronica Redaelli, Paolo Cappella, Pietro Zerbi, John Vincent Roughan, Luca Porcu, Davide Soranna, Laura Calvillo.

**Funding acquisition:** John Vincent Roughan, Laura Calvillo.

**Investigation:** Veronica Redaelli, Alice Bosi, Paolo Cappella, Daniele Di Lernia, John Vincent Roughan, Laura Calvillo.

**Methodology:** Veronica Redaelli, Alice Bosi, Fabio Luzi, Paolo Cappella, Pietro Zerbi, Nicola Ludwig, Daniele Di Lernia, John Vincent Roughan, Luca Porcu, Laura Calvillo.

**Project administration:** Laura Calvillo.

**Resources:** Fabio Luzi, Pietro Zerbi, Nicola Ludwig, John Vincent Roughan, Gianfranco Parati, Laura Calvillo.

**Software:** Veronica Redaelli, Nicola Ludwig.

**Supervision:** Gianfranco Parati, Laura Calvillo.

**Validation:** Daniele Di Lernia, John Vincent Roughan, Laura Calvillo.

**Visualization:** Gianfranco Parati, Laura Calvillo.

**Writing – original draft:** Laura Calvillo.

**Writing – review & editing:** Veronica Redaelli, Fabio Luzi, Paolo Cappella, Pietro Zerbi, Nicola Ludwig, Daniele Di Lernia, John Vincent Roughan, Luca Porcu, Davide Soranna, Gianfranco Parati, Laura Calvillo.

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
