## [Decision Letter · Decision Letter 0]

12 Apr 2021

PONE-D-21-03073

Neuroinflammation, body temperature and behavioural changes in CD1 Male Mice Undergoing Acute Restraint Stress: an Exploratory Study.

PLOS ONE

Dear Dr. Calvillo,

Thank you for submitting your manuscript to PLOS ONE. After careful consideration, we feel that it has merit but does not fully meet PLOS ONE’s publication criteria as it currently stands. Therefore, we invite you to submit a revised version of the manuscript that addresses the points raised during the review process.

Please refer to editorial recommendations by both reviewer 1 and 2.  In particular, addressing reviewer 1's concerns about a lack of discussion of confounding effects and the underpowered nature of the study are essential revisions to be made.  

We look forward to receiving your revised manuscript.

Kind regards,

Kimberly R. Byrnes, Ph.D.

Academic Editor

PLOS ONE

Journal Requirements:

Please include your tables as part of your main manuscript and remove the individual files. Please note that supplementary tables (should remain/ be uploaded) as separate "supporting information" files

Thank you for stating the following in the Competing Interests section:

The authors have declared that no competing interests exist.

We note that one or more of the authors are employed by a commercial company: Research & Services Dept, E.m.c2 srl, Varese, Italy.

3a, Please provide an amended Funding Statement declaring this commercial affiliation, as well as a statement regarding the Role of Funders in your study. If the funding organization did not play a role in the study design, data collection and analysis, decision to publish, or preparation of the manuscript and only provided financial support in the form of authors' salaries and/or research materials, please review your statements relating to the author contributions, and ensure you have specifically and accurately indicated the role(s) that these authors had in your study. You can update author roles in the Author Contributions section of the online submission form.

3b. Please also provide an updated Competing Interests Statement declaring this commercial affiliation along with any other relevant declarations relating to employment, consultancy, patents, products in development, or marketed products, etc. 

Reviewers' comments:

Reviewer's Responses to Questions

**Comments to the Author**

1. Is the manuscript technically sound, and do the data support the conclusions?

Reviewer #1: Partly

Reviewer #2: Yes

2. Has the statistical analysis been performed appropriately and rigorously? 

Reviewer #1: No

Reviewer #2: No

3. Have the authors made all data underlying the findings in their manuscript fully available?

Reviewer #1: Yes

Reviewer #2: Yes

4. Is the manuscript presented in an intelligible fashion and written in standard English?

Reviewer #1: No

Reviewer #2: Yes

5. Review Comments to the Author

Reviewer #1: This is a very interesting study to determine the impact of several factors common to animal handling that might impact study outcome. In particular, the authors focused on: 1. impact of bright light (500 lux) and 15 min restraint, procedures common to injections and type of handling (tail hold, tunnel or mechanoreceptive handling). While the overall intent of the study is laudatory, there are some concerns.

Unfortunately, the manuscript is underdeveloped. It is difficult to evaluate the study as it stands and require significant revisions before being properly evaluated. The experimental design is confusing - there are many factors that come into play and are not well controlled. The data presented does not show all the behavioral outcomes measured (1st after handling and 2nd after stress stimuli/handling). There is also no discussion of how each of the treatments (handling or stress) might have confounding effects.

Additional concerns are listed below:

1. The authors conducted the study using various non-parametric tests. More significantly, the authors used a power analysis centering on chemokine data to determine the number of replicates per treatment group. For this study, the authors should be using the behavioral outcomes to determine the number of replicates. It would appear from the figures and the raw data that the study lacks power. Figures 3, 4 and 8 all have very large error bars (not mentioned if this is standard deviation or error). In addition, the error bars in these figures are unusual - the upper and the lower error bars are different sizes. Its unclear how that can happen.

2. There is a lot of gaps in the methods. CXCL12 and CCL2 were quantified using an ELISA. What is the intra- and inter-assay CV? What is the sensitivity of the assay for each of the markers?

3. Immunohistochemistry - no controls were included in the figure. Was the antibody pre-absorbed? How was the antibody characterized? The low magnification image makes it difficult to evaluate whether the positive signals are real and not background noise - a 40X magnification is a minimum for this Iba1.

4. Elevated plus maze study. It is important to be sure that the baseline is at least 20-30% in the open arms compared to the overall (ie 60-90 s of 300 s). As it stands, the EPM data is not so valid.

5. More details are needed to the sequence of when the studies were conducted relative to each other.

6. There is no mention of when the studies are conducted during the light cycle and relative to zeitgeber (such as lights on or lights off). This is important as it affects the EPM data, body weight (is it right after main feeding?), nest building - affected by diurnal time etc.

7. The overall paper can be better develop and edited.

Reviewer #2: Redaelli and colleagues describe numerous physiological and behavioral effects of acute (15 min) restraint stress on mice that were handled using three different techniques, conventional tail handling, mechanoceptive handling and the non-invasive tunnel handling method. The authors provide a clear rationale for these studies, namely that the physiological and behavioral effects induced by routine procedural use of restraint under bright light conditions may induce variability in results of subsequent endpoints of interest and increase the number required per group due to this variability. Specifically, acute restraint stress reduced body temperature, produce a cessation in body weight gain, increased grooming behavior and decreased naturalistic behaviors in the home cage. No significant restraint induced changes in bone marrow chemokines, activated microglial number in the PVN and entire brain, or on nesting behavior. Furthermore, the authors demonstrate that tunnel handling mitigates the effects of the acute restraint stress/bright light conditions.

Overall the methods are clear and the interpretation of data is appropriate. The manuscript would be of interest to the general readership of PLOS One. There are some concerns/comments that the authors may wish to address.

Concerns/Comments

Regarding the statistical analysis, it is somewhat difficult to draw direct comparisons between the handling groups using non-parametric Mann Whitney within subjects’ evaluation of stress for each given handling technique. If you normalize the data and use Kruskal Wallis, you may wish to compute the effect size for Kruskal-Wallis test as the eta squared based on the H-statistic. That effect size will be a more objective method to compare between handling groups.

In the introduction, the authors suggest that chronic restraint is a well-established model to produce post-traumatic stress disorder or model depression. It is not really appropriate to say it produces PTSD, rather that it may induce behaviors that have relevance to the clinical conditions of PTSD and major depressive disorder.

Only male animals are used in this study. Is there evidence in the literature that show that females respond similarly to males in terms of stress responsivity following tunnel handling? It would be appropriate to add to the discussion to address the issue of sex as a biological variable. Useful reference include Sensini et al2020, DOI: 10.1038/s41598-020-74279-3 and Gouveia and Hurst 2019, DOI: 10.1038/s41598-019-56860-7, which is already referenced in this manuscript.

It may be useful to measure peripheral corticosterone levels in these animals to use as another physiological measure. I would expect that your tunnel handled animals may not have as high levels. Another control would be tail handled animals that are intermittently but repeatedly exposed to the restraint device. Would there be habituation of the behavioral effects observed in such animals.

Methods

• It would be helpful to explicitly state how the animals were transferred to the experimental apparatus. I assume sentinels were transferred by tail handling.

• With regards to the use of sentinels as control animals, were these animals exposed to other pathogens etc. in the room? Are they housed specifically on the same rack as the experimental animals?

• To improve the flow of the manuscript, handling methods should come directly after husbandry.

• Move the following information from the experimental design to the statistical section – “The numerosity within the groups derived from a power calculation for CXCL12 chemokine concentration in the BM (software available at http://www.biomath.info/). On the base of the literature, a standard deviation of 50 and a difference of 70 pg/ml was expected between groups to have an alpha value less than 0,05 and a power of 0,80.”

• For the nesting procedure, 30 seconds is a very short duration in which to observe construction of a nest. Were the additional scores observed at later time points? There may have been a delay in the onset of nesting building, and this would be more informative than a snap shot just after nesting material presentation or 24 h totals of nesting. I would draw your attention to newer publications that utilize nesting as a behavioral endpoint, for example Jacobson 2020 DOI: 10.1016/j.neuropharm.2020.108254. In that study, nesting was evaluated over a 5 hour period. Also, please clarify in the methods whether the statistical analysis was conducted on nest score at 24 h.

• For body weight, data is represented in grams. It perhaps may be most useful to have a normalized value of weight gain/weight loss. Using the percentage of body weight is a useful transformation for statistical analysis between groups.

• It is always difficult to interpret repeated assays of anxiety measures. It is most common that upon repeated testing animals tend to explore more as the aversive quality of the neophobia is diminished. This might explain some of the variance in the post stress time spent in the open arms. Have you considered transforming the data as ratio of post/pre stress?

Results/Figures

• For the data which was analyzed with Kruskal Wallis, please provide the H values.

• Provide the actual values for nesting in the results section and also in Figure 2. The photographs of the nests are appropriate, but the data should be included in that figure.

• Figure 5 - please place the appropriate symbols that indicate the significant effects of stress/handling on grooming and other parameters.

• Figure 10 - only one photomicrograph of Iba1+ immunoreactivity is provided. Please provide the data obtained from all groups and a representative photomicrograph for each handling group.

6. PLOS authors have the option to publish the peer review history of their article (what does this mean?). If published, this will include your full peer review and any attached files.

Reviewer #1: No

Reviewer #2: No

---

## [Author Response · Author response to Decision Letter 0]

7 Jun 2021

Response to Reviewers

1. Is the manuscript technically sound, and do the data support the conclusions?

Reviewer #1: Partly

Reviewer #2: Yes

2. Has the statistical analysis been performed appropriately and rigorously?

Reviewer #1: No

Reviewer #2: No

3. Have the authors made all data underlying the findings in their manuscript fully available?

Reviewer #1: Yes

Reviewer #2: Yes

4. Is the manuscript presented in an intelligible fashion and written in standard English?

Reviewer #1: No

Reviewer #2: Yes

5. Review Comments to the Author

Reviewer #1: This is a very interesting study to determine the impact of several factors common to animal handling that might impact study outcome. In particular, the authors focused on: 1. impact of bright light (500 lux) and 15 min restraint, procedures common to injections and type of handling (tail hold, tunnel or mechanoreceptive handling). While the overall intent of the study is laudatory, there are some concerns.

Unfortunately, the manuscript is underdeveloped. It is difficult to evaluate the study as it stands and require significant revisions before being properly evaluated. The experimental design is confusing - there are many factors that come into play and are not well controlled. 

We thank Reviewer for this comment which has allowed us to improve the experimental design description. A revised figure 1 and a more detailed explanation of our study design is now included in the manuscript (Experimental Design paragraph).

The data presented does not show all the behavioral outcomes measured (1st after handling and 2nd after stress stimuli/handling). 

We thank the Reviewer for this observation. Following the reviewer’s suggestion we have now included a new table describing the behavioural outcomes measured and not displayed in the original manuscript (1st after handling and 2nd after stress stimuli/handling) (Tab. 3). Moreover, another table with immunohistochemistry values has been added (Tab. 1). 

There is also no discussion of how each of the treatments (handling or stress) might have confounding effects.

We thank the Reviewer for this comment which was probably derived from an insufficiently clear description of the experimental design in our original manuscript. While handling was the study treatment factor, which was thus different among the three experimental groups, R&L stress was not a treatment, but rather the common condition which all treated mice were exposed to. Therefore, stress cannot have generated confounding effects, as it was administered in the same way to all experimental groups. Our study was aimed at assessing the occurrence of systematic differences among groups (all exposed to the same R&L stress), in response to a given specific treatment (handling). We have to thank the reviewer for this comment, which has allowed us to better describe the flow chart of the experimental design of our work, now included in the revised manuscript.

Additional concerns are listed below:

1. The authors conducted the study using various non-parametric tests. More significantly, the authors used a power analysis centering on chemokine data to determine the number of replicates per treatment group. For this study, the authors should be using the behavioral outcomes to determine the number of replicates. It would appear from the figures and the raw data that the study lacks power. 

This reviewer’s comment is theoretically important, as it addresses a basic and fundamental aspect of any experimental study design, but, also in this case, it was probably originated by an insufficiently clear description of our primary endpoint in the original manuscript. The primary endpoint of this study was to explore a possible neuroinflammatory effect of acute restraint under bright light. This was the endpoint for which we received the grant NC/S000887/1 provided to by the UK NC3R’s. Assessing response to behavioural tests was not our main goal, but rather a secondary aim of our study, collected to obtain a more comprehensive assessment of animals’ response to stress. Thus, study power calculation was performed on the primary endpoint, i.e. on the CXCL12 chemokine content in the bone-marrow. The sample size determined through this power calculation came out to be adequate also to observe a significant change in the responses to two behavioural tests, EPM and HCS. We have now better clarified these issues in the revised paper. 

In fact, exploring possible stress-related neuroinflammation was a fundamental point in our investigation, since Ataka and colleagues [1] revealed an important increase in bone marrow-derived microglia infiltration in the brain and a reduced stromal cell-derived factor-1 (SDF-1 or CXCL12) content in the bone marrow after chronic psychological stress. Being bone marrow-derived microglia infiltration in the brain related with neurogenic hypertension [2], it was a fundamental point to explore whether also common R&L stress might induce neuroinflammation, thus becoming a possible source of bias. 

We applied non-parametric tests since in our intention this was to be an exploratory experiment useful to provide information for subsequent research in our laboratory, and we wanted to be conservative. Another intention we had in submitting this preliminary observation, was to disseminate the information of a possible risk of bias when a common procedure like restraint has to be used. 

We wish to thank the reviewer for these comments which have allowed us to clarify in our revised paper our approach to study power analysis and the choice of the primary and secondary endpoints (see Statistical analysis paragraph and Introduction). 

Figures 3, 4 and 8 all have very large error bars (not mentioned if this is standard deviation or error). In addition, the error bars in these figures are unusual - the upper and the lower error bars are different sizes. It’s unclear how that can happen.

Thank you for this comment. We have now deleted the error bars which represented the interquartile ranges and not the standard deviations. We have now also specified that the data showed in the figures were median values.

2. There is a lot of gaps in the methods. CXCL12 and CCL2 were quantified using an ELISA. What is the intra- and inter-assay CV? What is the sensitivity of the assay for each of the markers?

We thank the Reviewer for this comment which allowed to improve the Methods section. The required details are now in the revised manuscript. 

3. Immunohistochemistry - no controls were included in the figure. Was the antibody pre-absorbed? How was the antibody characterized? The low magnification image makes it difficult to evaluate whether the positive signals are real and not background noise - a 40X magnification is a minimum for this Iba1.

Immunohistochemistry was performed according to manufacturer instructions. The section was pre-treated using heat mediated antigen retrieval with Ethylenediaminetetraacetic acid (EDTA) for 3 cycles of 5 minutes each in microwave oven. The section was then incubated with recombinant Iba-1 antibody, 1/2000 dilution, for 2 hours at room temperature. Abcam Iba-1 is a recombinant, non-pre-absorbed antibody. In order to obtain a better result, we used the MACH4 HRP polymer, a very effective amplification system, as a secondary antibody. DAB was used as the chromogen. The section was then counterstained with haematoxylin and mounted. These details are now in the manuscript.

In our hands, immunohistochemistry worked very well, with only optimization of the retrieval conditions. We agree with the reviewer that higher magnification (40x) can better show the clean background and specificity of the signal. As stated by many authors, the positive controls are microglial cells in the rat brain: an image at high magnification (40x) will allow to make evident the well delineated signal in the cell body and in the dendrites. Thanks to the Reviewer’s suggestion, the methods section has been corrected accordingly and an updated immunohistochemistry image (now Fig. 6) is now present in the manuscript. 

4. Elevated plus maze study. It is important to be sure that the baseline is at least 20-30% in the open arms compared to the overall (ie 60-90 s of 300 s). As it stands, the EPM data is not so valid.

We thank the Reviewer for this comment which allows to clarify an important point. According to Walf et al [7], we think that it is not possible to be ‘sure’ of anything with regard to the proportion of time mice will spend on the open arms at baseline versus after any stress/anxiety-causing procedure. In fact, this important paper describes some of the limitations of the test procedure, suggesting ideal conditions for its performance and giving example data. These authors describe how baseline open arm time can be confounded by mice ‘freezing’ on the open arms, i.e. if this is around 30% of the overall test time. Therefore, the time the reviewer suggests might actually represent a confounding effect rather than be an important value to be sure of. This was not the case with any of our baseline results and we consider this an advantage. In one study on anxiolytic treatment that Walf et al cite, they describe how, at baseline (control condition), 2% of time was spent on the open arms by mice in a 300s trial, whereas anxiolytic treated mice spent 7% time on the open arms, and this was deemed a significant difference. Therefore, we respectfully do not consider our EPM data to be not so valid. 

5. More details are needed to the sequence of when the studies were conducted relative to each other.

We thank the Reviewer for this suggestion. A more detailed explanation on the sequence of the studies conducted relative to each other is now in the manuscript in the “experimental design” chapter as well as in the new Fig. 1 

6. There is no mention of when the studies are conducted during the light cycle and relative to zeitgeber (such as lights on or lights off). This is important as it affects the EPM data, body weight (is it right after main feeding?), nest building - affected by diurnal time etc.

We thank the Reviewer for the comment. We agree that these aspects are all affected by the time during the day when the testing is done. However, we wished to perform a comparative investigation of the impact of the stress procedure as it is normally applied – ie during the day. We have acknowledged this potential limitation in the new Study Limitation paragraph and cited Walf and Frye [7]. The clarification about light cycle is also in the revised manuscript in the “experimental design” chapter.

7. The overall paper can be better developed and edited.

We have tried to improve our manuscript as suggested by the Reviewer. 

Reviewer #2: Redaelli and colleagues describe numerous physiological and behavioral effects of acute (15 min) restraint stress on mice that were handled using three different techniques, conventional tail handling, mechanoceptive handling and the non-invasive tunnel handling method. The authors provide a clear rationale for these studies, namely that the physiological and behavioral effects induced by routine procedural use of restraint under bright light conditions may induce variability in results of subsequent endpoints of interest and increase the number required per group due to this variability. Specifically, acute restraint stress reduced body temperature, produce a cessation in body weight gain, increased grooming behavior and decreased naturalistic behaviors in the home cage. No significant restraint induced changes in bone marrow chemokines, activated microglial number in the PVN and entire brain, or on nesting behavior. Furthermore, the authors demonstrate that tunnel handling mitigates the effects of the acute restraint stress/bright light conditions. Overall the methods are clear and the interpretation of data is appropriate. The manuscript would be of interest to the general readership of PLOS One. 

We thank Reviewer for this comment.

There are some concerns/comments that the authors may wish to address.

Concerns/Comments

Regarding the statistical analysis, it is somewhat difficult to draw direct comparisons between the handling groups using non-parametric Mann Whitney within subjects’ evaluation of stress for each given handling technique. If you normalize the data and use Kruskal Wallis, you may wish to compute the effect size for Kruskal-Wallis test as the eta squared based on the H-statistic. That effect size will be a more objective method to compare between handling groups. 

We thank the Reviewer for this suggestion, now we included in the results section the eta-square based on H statistic and we added an explanation of this index in statistical analysis section.

In the introduction, the authors suggest that chronic restraint is a well-established model to produce post-traumatic stress disorder or model depression. It is not really appropriate to say it produces PTSD, rather that it may induce behaviors that have relevance to the clinical conditions of PTSD and major depressive disorder. 

We thank the Reviewer for this clarification. The text is now corrected accordingly.

Only male animals are used in this study. Is there evidence in the literature that show that females respond similarly to males in terms of stress responsivity following tunnel handling? It would be appropriate to add to the discussion to address the issue of sex as a biological variable. Useful reference includes Sensini et al2020, DOI: 10.1038/s41598-020-74279-3 and Gouveia and Hurst 2019, DOI: 10.1038/s41598-019-56860-7, which is already referenced in this manuscript. 

We thank the Reviewer for this suggestion. The original article by Hurst and West [8] showed that both males and females of three different strains responded well (i.e., had lowered anxiety in response) to tunnel handling, and there was a slight tendency for females to have even lower anxiety than males. A more recent study has replicated this finding but found that male mice may be more negatively impacted by tail handling than females. This did not drive our selection of a male-only design, which was based on the fact that, since gender can influence response to handling [9-10] and considering the exploratory nature of the study, we had to reduce possible confounding effects, avoiding to introduce this biological variable. Nevertheless, our choice to use males was a fortunate one as these may be more susceptible to the aversive tail handling than females [10]. A clarification about the issue of sex as a biological variable is now added in the Experimental Design and in the Discussion chapters.

It may be useful to measure peripheral corticosterone levels in these animals to use as another physiological measure. I would expect that your tunnel handled animals may not have as high levels. 

These are all going point worth remembering in the design of future studies. Thank you for the suggestion. Nevertheless, while corticosterone assays can provide useful information with regard to the stressful consequences of procedures, the success of both of the commonest methods of sampling (plasma or faecal) are tightly bound to the timing of collection. In summary, there is some evidence for both positive and negative impacts on either faecal or plasma corticosterone (please see the summary table NC3Rs has produced at: https://www.nc3rs.org.uk/mouse-handling-research-papers). Moreover, in the present work it was not possible to safely withdraw plasma because the blood collection procedure might have influenced neuroinflammatory response, that was our primary endpoint. In fact, in order to have sufficient quantity of sample we should have performed a cardiac puncture, which has been demonstrated to cause a cytokine elevation with respect to collection of blood from the facial vein, which has a less impact on cytokines level, but does not allow to take the necessary amount of blood [11]. This clarification is now added in the new section “Study Limitations”

Another control would be tail handled animals that are intermittently but repeatedly exposed to the restraint device. Would there be habituation of the behavioral effects observed in such animals.

Again, we thank for this suggestion. Our primary goals were to verify possible stress-related alteration in neuroinflammation and in BAT temperature, caused by the common setting of restraint animals for injections or similar procedures. The approach suggested by the reviewer is certainly of interest. However, in our study we decided to consider as controls animals not exposed to any restraint procedure. In our future studies we will consider adding this additional control, and we are grateful for the advice.

Methods

• It would be helpful to explicitly state how the animals were transferred to the experimental apparatus. I assume sentinels were transferred by tail handling.

The reviewer’s comment is very appropriate: sentinels did not undergo any behavioural test, as specified in the first lines of experimental design chapter, nevertheless when necessary for cage cleaning, they were transferred by tail handling. This clarification is now added in the text.

• With regards to the use of sentinels as control animals, were these animals exposed to other pathogens etc. in the room? Are they housed specifically on the same rack as the experimental animals?

We thank the Reviewer for this question, sentinels were exposed to the same environment the others mice were exposed to and were housed specifically on the same rack as the experimental animals. This detail is now added in the experimental design chapter.

• To improve the flow of the manuscript, handling methods should come directly after husbandry.

We thank the Reviewer for this suggestion, the chapter has been moved accordingly.

• Move the following information from the experimental design to the statistical section – “The numerosity within the groups derived from a power calculation for CXCL12 chemokine concentration in the BM (software available at http://www.biomath.info/). On the base of the literature, a standard deviation of 50 and a difference of 70 pg/ml was expected between groups to have an alpha value less than 0,05 and a power of 0,80.”

We thank the Reviewer for this suggestion, this part is now in the statistical section.

• For the nesting procedure, 30 seconds is a very short duration in which to observe construction of a nest. Were the additional scores observed at later time points? There may have been a delay in the onset of nesting building, and this would be more informative than a snap shot just after nesting material presentation or 24 h totals of nesting. I would draw your attention to newer publications that utilize nesting as a behavioral endpoint, for example Jacobson 2020 DOI: 10.1016/j.neuropharm.2020.108254. In that study, nesting was evaluated over a 5 hour period. Also, please clarify in the methods whether the statistical analysis was conducted on nest score at 24 h.

Thirty seconds of footage was used to allow us to score the completed nest not its construction. Our description of the scoring procedure was obviously misleading and has been corrected, the revised manuscript now clarifies this.

We are also grateful for bringing this article to our attention. Unfortunately, this interesting work was published after completion of our experiments. Nevertheless, we took care to repeat the observations at the same time, around 7 am each day, and to follow the method described by Hess [12] who first published the nest score procedure. In our future studies, the suggested procedure of Jacobson will certainly be taken into consideration. The required clarification on the statistical analysis conducted on nest score at 24 h is now present in the revised manuscript.

• For body weight, data is represented in grams. It perhaps may be most useful to have a normalized value of weight gain/weight loss. Using the percentage of body weight is a useful transformation for statistical analysis between groups.

We are grateful for this suggestion. A new table and a new figure displaying the percentage of body weight increased is now in the revised manuscript (Tab. 2, Fig. 7).

• It is always difficult to interpret repeated assays of anxiety measures. It is most common that upon repeated testing animals tend to explore more as the aversive quality of the neophobia is diminished. This might explain some of the variance in the post stress time spent in the open arms. Have you considered transforming the data as ratio of post/pre stress?

We are grateful for this suggestion which we have followed by adding a new table (Tab.3) that shows all the EPM outcome measured as well as a ratio of post/pre-stress time spent in the open arms. Regarding the difficulty to interpret repeated assays of anxiety measures, it is true that the EPM is susceptible to variation if the test is applied successively, unless there are several weeks passing between the before and after conditions. If EPM had been our main endpoint, the design would have obviously to be different, measuring open-field responding first, then test using the EPM in one half of the mice, then the reverse in the other. Following this, we could have looked at proportional changes in anxiety-like responding in a more balanced manner while equalising the exposure of the mice in each treatment group to the novel environment each method poses. Again, this suggestion is extremely useful for our future studies. 

The reason of adding EPM as further behavioural test, was to have a complete overview within this exploratory study, despite behavioural tests were not the primary endpoints of the project. We have now added this latter point in the text (Introduction). 

Results/Figures

• For the data which was analyzed with Kruskal Wallis, please provide the H values.

Thanks for the suggestion, H values are now provided.

• Provide the actual values for nesting in the results section and also in Figure 2. The photographs of the nests are appropriate, but the data should be included in that figure.

Reviewer’s comment is very useful, results section and figure 2 are now corrected accordingly.

• Figure 5 - please place the appropriate symbols that indicate the significant effects of stress/handling on grooming and other parameters.

Thanks for the suggestion, an appropriate description of the significant effects of stress is now in the figure and in the legend.

• Figure 10 - only one photomicrograph of Iba1+ immunoreactivity is provided. Please provide the data obtained from all groups and a representative photomicrograph for each handling group.

We thank the Reviewer for this suggestion. A corrected figure representative for each handling group is now present, as well as a new table with the data described in immunohistochemistry results chapter and a series of images with different degrees of magnification (2x, 20x and 40x). 

Reference

1. Ataka K, Asakawa A, Nagaishi K, Kaimoto K, Sawada A, Hayakawa Y, et al. Bone marrow-derived microglia infiltrate into the paraventricular nucleus of chronic psychological stress-loaded mice. PLoS One. 2013;8(11):1–14. 

2. Calvillo, L. Gironacci, M.M. Crotti, L. Meroni, P.L. Parati G. Neuroimmune crosstalk in the pathophysiology of hypertension. Nat Rev Cardiol. 2019;16:476-490. 

3. Redaelli, V., Papa, S. Marsella, G. Grignaschi, G., Bosi, A. Ludwig, N. Luzi, F., Vismara, I., Rimondo, S. Veglianese, P. Tepteva, S. Mazzola, S. Zerbi, P. Porcu, L. Roughan, J.V. Parati, G. and Calvillo L. A Refinement approach in a mouse model of rehabilitation research. Analgesia strategy, Reduction approach and infrared thermography in spinal cord injury. PLoS One. 2019;14:e0224337. 

4. Devlin MJ. The “skinny” on brown fat, obesity, and bone. Am J Phys Anthropol. 2015;156(S59):98–115. 

5. Stanford KI, Middelbeek RJW, Townsend KL, An D, Nygaard EB, Hitchcox KM, et al. Brown adipose tissue regulates glucose homeostasis and insulin sensitivity. J Clin Invest. 2013;123(1):215–23. 

6. Fornasier M., Redaelli V., Tarantino A., Luzi F. VM. Infrared thermography (IRT) in nude mice: an alternative method for body temperature measurement. In: Atti Scand FELASA 2010, Helsinki, June 14-17, 2010. 2010. 

7. Walf A., Frye C. The use of the elevated plus maze as an assay of anxiety-related behavior in rodents. Nat Protoc. 2007;2(2):322–8. 

8. Hurst JL, West RS. Taming anxiety in laboratory mice. Nat Methods [Internet]. 2010;7(10):825–6. Available from: http://dx.doi.org/10.1038/nmeth.1500

9. Gouveia K, Hurst JL. Improving the practicality of using non-aversive handling methods to reduce background stress and anxiety in laboratory mice. Sci Rep. 2019;9(1):1–19. 

10. Sensini F, Inta D, Palme R, Brandwein C, Pfeiffer N, Riva MA, et al. The impact of handling technique and handling frequency on laboratory mouse welfare is sex-specific. Sci Rep [Internet]. 2020;10(1):1–9. Available from: https://doi.org/10.1038/s41598-020-74279-3

11. Mella JR, Chiswick EL, King E, Remick DG. Location, location, location: Cytokine concentrations are dependent on blood sampling site. Shock. 2014;42(4):337–42. 

12. Hess SE, Rohr S, Dufour BD, Gaskill BN, Pajor EA, Garner JP. Home improvement: C57BL/6J mice given more naturalistic nesting materials build better nests. J Am Assoc Lab Anim Sci [Internet]. 2008;47(6):25–31. Available from: http://www.pubmedcentral.nih.gov/articlerender.fcgi?artid=2687128&tool=pmcentrez&rendertype=abstract

---

## [Decision Letter · Decision Letter 1]

25 Jun 2021

PONE-D-21-03073R1

Neuroinflammation, body temperature and behavioural changes in CD1 Male Mice Undergoing Acute Restraint Stress: an Exploratory Study.

PLOS ONE

Dear Dr. Calvillo,

Thank you for submitting your manuscript to PLOS ONE. After careful consideration, we feel that it has merit but does not fully meet PLOS ONE’s publication criteria as it currently stands. Therefore, we invite you to submit a revised version of the manuscript that addresses the points raised during the review process.

Please pay particular attention to the request of reviewer 1 to provide additional controls for data analysis and interpretation.

We look forward to receiving your revised manuscript.

Kind regards,

Kimberly R. Byrnes, Ph.D.

Academic Editor

PLOS ONE

Reviewers' comments:

Reviewer's Responses to Questions

**Comments to the Author**

1. If the authors have adequately addressed your comments raised in a previous round of review and you feel that this manuscript is now acceptable for publication, you may indicate that here to bypass the “Comments to the Author” section, enter your conflict of interest statement in the “Confidential to Editor” section, and submit your "Accept" recommendation.

Reviewer #1: (No Response)

Reviewer #2: (No Response)

2. Is the manuscript technically sound, and do the data support the conclusions?

Reviewer #1: Partly

Reviewer #2: Yes

3. Has the statistical analysis been performed appropriately and rigorously? 

Reviewer #1: Yes

Reviewer #2: Yes

4. Have the authors made all data underlying the findings in their manuscript fully available?

Reviewer #1: No

Reviewer #2: Yes

5. Is the manuscript presented in an intelligible fashion and written in standard English?

Reviewer #1: Yes

Reviewer #2: No

6. Review Comments to the Author

Reviewer #1: I thank the authors for the improved manuscript. Figure 1 is helpful in interpreting the experiment conducted.

There are still a few points of concern:

1. NEW. It is unclear when the behavioral tests and samples were collected during the light phase of the day (0600 - 1800 h). Is it consistent? It makes a difference if it is close to the transitions (ie 0600 - ~ 0800 h and 1500 - ~1800 h) for the mice. Since this is a NC3R study, the authors should also state what time of the day the vivarium caretakers enter the room as this can also have an impact on the study.

2. The authors made a very good counter argument about the very low baseline time in open arms (significantly less than 20-30%). However, the data does not reflect their argument. There should be parallel changes in open arm time and open arm entry. I am okay to waive the 20-30% baseline but the authors should show that the mice were actively exploring. In addition to the measures in Table 3A, it would be good to include activity in the open arm and activity in the closed arm. Other measures of exploration are head dip, sniffing etc.

3. Immunohistochemistry.

The 40X is an improvement. Thank you.

Regarding controls for the Iba1+ cells, the authors still did not address what controls they used. Controls for immunohistochemistry typically includes: 1. removal of primary antibody, 2. preabsorption of the antibody using a peptide (this should be easy to purchase or synthesize) or 3. use a Iba1 knockout mouse. The third option is idea but not always possible. Option #2 is the more commonly used option. The reason it is necessary to conduct a negative control is that immunohistochemistry is sensitive to many subtle changes in protocol - fixative, antigen retrieval etc etc etc. These subtle changes can cause changes to the epitope which can lead to false positives or partial positives/negatives.

4. NEW. More details are needed for where Iba1+ cells were counted. More details are needed to define the PVN and the entire section posterior to the midpoint of the brain. More typical details include: 1. using a stereotaxic atlas to provide the coordinates (rostro-caudal and lateral) for what you are looking at and 2. a photomicrograph of the region of interest that is outlined. In addition, more information is also needed to how the authors measured the Iba+ cells in a non-biased manner.

5. NEW. In the response to reviewers, the authors state that "an image at high magnification (40X) will allow to make evident the well delineated signal in the cell body and in the dendrites." Microglia do not have dendrites.

Reviewer #2: Please ensure that the term sex is used throughout the manuscript not sex. Your mice have a sex, they do not have a gender.

Tables & Figures – with regard to presentation of the data. It would be best to present the data for the sentinels (true controls) in the first line.

If the data are presented as a figure it is not necessary to present a table also, for example Table 3b contains the same information as presented in figure 8.

Table 2 – data from TH animals were not included here, so perhaps you may wish to omit this line from the table to avoid confusion.

Figures 3 and 4 should be combined into one Figure.

The data for the nesting behavior has not been added in table or figure format.

For the EPM it is not conventional to use the term fold increase. Rather choose a better descriptor, such as ratio of post/pre, or normalized to prestress baseline.

There remains a number of grammatical and typographical errors. The manuscript would benefit from editing by a native English speaker.

7. PLOS authors have the option to publish the peer review history of their article (what does this mean?). If published, this will include your full peer review and any attached files.

Reviewer #1: No

Reviewer #2: No

---

## [Author Response · Author response to Decision Letter 1]

9 Oct 2021

Please pay particular attention to the request of reviewer 1 to provide additional controls for data analysis and interpretation.

Reviewer's Responses to Questions

Comments to the Author

1. If the authors have adequately addressed your comments raised in a previous round of review and you feel that this manuscript is now acceptable for publication, you may indicate that here to bypass the “Comments to the Author” section, enter your conflict of interest statement in the “Confidential to Editor” section, and submit your "Accept" recommendation.

Reviewer #1: (No Response)

Reviewer #2: (No Response)

2. Is the manuscript technically sound, and do the data support the conclusions?

Reviewer #1: Partly

Reviewer #2: Yes

3. Has the statistical analysis been performed appropriately and rigorously?

Reviewer #1: Yes

Reviewer #2: Yes

4. Have the authors made all data underlying the findings in their manuscript fully available?

Reviewer #1: No

Reviewer #2: Yes

5. Is the manuscript presented in an intelligible fashion and written in standard English?

Reviewer #1: Yes

Reviewer #2: No

6. Review Comments to the Author

Reviewer #1: I thank the authors for the improved manuscript. Figure 1 is helpful in interpreting the experiment conducted.

There are still a few points of concern:

1. NEW. It is unclear when the behavioral tests and samples were collected during the light phase of the day (0600 - 1800 h). Is it consistent? It makes a difference if it is close to the transitions (ie 0600 - ~ 0800 h and 1500 - ~1800 h) for the mice. Since this is a NC3R study, the authors should also state what time of the day the vivarium caretakers enter the room as this can also have an impact on the study.

We thank the reviewer for highlighting this important detail. The time when the behavioral tests and samples were collected was consistently during the light phase, at least one hour away from the transition phase. The specific time of the day is reported in the “Experimental Design” paragraph, where the procedures performed in Day 2, 4 and 5 were described. In the vivarium, light was on at 6 am and off at 6pm. When necessary, caretakers entered the room always at the same hour which was around 9 am. This information is now in the manuscript in the “Experimental Design” paragraph (from line 214 of the Revised Manuscript with Track Changes).

2. The authors made a very good counter argument about the very low baseline time in open arms (significantly less than 20-30%). However, the data does not reflect their argument. There should be parallel changes in open arm time and open arm entry. I am okay to waive the 20-30% baseline but the authors should show that the mice were actively exploring. In addition to the measures in Table 3A, it would be good to include activity in the open arm and activity in the closed arm. Other measures of exploration are head dip, sniffing etc.

Thank you for acknowledging the validity of the counter argument in our initial reply. We agree that there should be an approximate correlation between the number of open arm entries and time spent on the open arms of the EPM. We cannot explain such low numbers of open arm entries but note that it is not unusual. For example, Henderson et al. reported an average of less than 1 open arm entry following tail-handling, increasing to only 3.5 in mice handled non-aversively (using a tunnel: https://doi.org/10.1038/s41598-020-71476-y). In their study the corresponding percentage of time mice in each handling group spent on the open arms was only 7 and 18%, respectively. In our study, ignoring the relatively minor effect that handling method had, the equivalent values were 5.6% before stress vs. 7.6 afterwards. We also checked to see if our data met the reviewers correct assertion that there should be parallel changes in open arm entries and open arm time. The proportion of entries vs. time was between 3-4 before handling and between 5 and 7 afterwards. We admit that these values are still low, but actually do parallel one another. Furthermore, and as we previously said, the behavioural outcomes were subsidiary to our main objective of exploring a possible neuroinflammatory effect of acute restraint under bright light. Handling method, and whether it might ameliorate the response to this stressor was secondary. Since there were not significant effects of this on the other parameter we evaluated (protected-stretched attend), we do not think it would be a sensible use of time to re-evaluate the activity of the mice on the open and closed arms, as we would likely arrive at the same conclusion – that handling had minimal effect. That’s said we do agree with the reviewer that these values are low, and have acknowledged this in the “Study Limitation” paragraph (from line 609 of the Revised Manuscript with Track Changes). 

3. Immunohistochemistry.

The 40X is an improvement. Thank you.

Regarding controls for the Iba1+ cells, the authors still did not address what controls they used. Controls for immunohistochemistry typically includes: 1. removal of primary antibody, 2. preabsorption of the antibody using a peptide (this should be easy to purchase or synthesize) or 3. use an Iba1 knockout mouse. The third option is idea but not always possible. Option #2 is the more commonly used option. The reason it is necessary to conduct a negative control is that immunohistochemistry is sensitive to many subtle changes in protocol - fixative, antigen retrieval etc etc etc. These subtle changes can cause changes to the epitope which can lead to false positives or partial positives/negatives.

Thank you for acknowledging the improvement of the 40X figure. In order to have controls for immunohistochemistry we repeated reactions on a double set of slides of all brain samples, simultaneously, with Iba1 antibody and without it. The reactions on slides performed by removal of primary antibody did not show any signal. We have now specified the negative control used in the text (line 368) and modified figure 5 accordingly.

4. NEW. More details are needed for where Iba1+ cells were counted. More details are needed to define the PVN and the entire section posterior to the midpoint of the brain. More typical details include: 1. using a stereotaxic atlas to provide the coordinates (rostro-caudal and lateral) for what you are looking at and 2. a photomicrograph of the region of interest that is outlined. In addition, more information is also needed to how the authors measured the Iba+ cells in a non-biased manner.

Thanks for the comment. We used the following atlas: Paxinos G. and Franklin, K.B.J., The mouse brain in stereotaxic coordinates, Academic Press, New York, 2001, ISBN 0-12-547637-X, in order to define the brain areas. The entire brain section posterior to the midpoint was taken 2 mm posterior to the bregma. According to Paxinos Atlas, we identify PVN just beneath the 3rd ventricle, at the following coordinates: -1.5 rostro-caudal; +/- 0.1 lateral (i.e. millimeters from bregma and from the midline, respectively). Atlas information and coordinates are now provided in the text (from line 372) and the photomicrograph of the region of interest that is outlined (PVN) is now in the modified fig. 5. Regarding information to how we measured the Iba+ cells in a non-biased manner, we provided a detailed description of our use of ImageJ software in the method paragraph of our original submission (from line 372); in particular, we specified that signals were automatically counted. Nevertheless, Reviewer’s comment is important and we have now added a sentence accordingly, also specifying that researchers performing histology were blinded to the treatment (from line 379). 

5. NEW. In the response to reviewers, the authors state that "an image at high magnification (40X) will allow to make evident the well delineated signal in the cell body and in the dendrites." Microglia do not have dendrites.

The Reviewer is right, thanks for highlighting this error in our previous response. 

Reviewer #2: Please ensure that the term sex is used throughout the manuscript not sex. Your mice have a sex, they do not have a gender.

Thanks for highlighting this error, which we have corrected accordingly.

Tables & Figures – with regard to presentation of the data. It would be best to present the data for the sentinels (true controls) in the first line.

Thank you for the suggestion. We have now modified all tables and figures accordingly.

If the data are presented as a figure it is not necessary to present a table also, for example Table 3b contains the same information as presented in figure 8.

We thank for this comment; nevertheless, a complete presentation with both table and figure for the EPM parameters was necessary to address the Reviewer 1’s comments.

Table 2 – data from TH animals were not included here, so perhaps you may wish to omit this line from the table to avoid confusion.

The meaning of these values was that there were no differences. Omitting these values would be in our opinion more and not less confusing. To address the reviewer’s concern, however, we have tried to clarify this issue in the legend. Thank you for the suggestion.

Figures 3 and 4 should be combined into one Figure.

Thank you for this suggestion. We have now combined Figure 3 and figure 4, accordingly.

The data for the nesting behavior has not been added in table or figure format.

Thank you for the suggestion. A new table 5 with the data for the nesting behaviour is now present in the manuscript

For the EPM it is not conventional to use the term fold increase. Rather choose a better descriptor, such as ratio of post/pre, or normalized to prestress baseline.

We thank the Reviewer for this comment. The table 3b and the EPM figure actually showed a ratio of post/pre, and we have double checked this by repeating all calculations. We have now edited figures, tables and manuscript text accordingly.

There remains a number of grammatical and typographical errors. The manuscript would benefit from editing by a native English speaker.

Thank you for the comment, the native English speaker co-author Dr. Roughan has edited the manuscript.

7. PLOS authors have the option to publish the peer review history of their article (what does this mean?). If published, this will include your full peer review and any attached files.

Authors are agreed to publish the peer review history of their article

Do you want your identity to be public for this peer review? For information about this choice, including consent withdrawal, please see our Privacy Policy.

Reviewer #1: No

Reviewer #2: No

---

## [Decision Letter · Decision Letter 2]

2 Nov 2021

Neuroinflammation, body temperature and behavioural changes in CD1 Male Mice Undergoing Acute Restraint Stress: an Exploratory Study.

PONE-D-21-03073R2

Dear Dr. Calvillo,

We’re pleased to inform you that your manuscript has been judged scientifically suitable for publication and will be formally accepted for publication once it meets all outstanding technical requirements.

Kind regards,

Kimberly R. Byrnes, Ph.D.

Academic Editor

PLOS ONE

Additional Editor Comments (optional):

Reviewers' comments:

Reviewer's Responses to Questions

**Comments to the Author**

1. If the authors have adequately addressed your comments raised in a previous round of review and you feel that this manuscript is now acceptable for publication, you may indicate that here to bypass the “Comments to the Author” section, enter your conflict of interest statement in the “Confidential to Editor” section, and submit your "Accept" recommendation.

Reviewer #2: All comments have been addressed

2. Is the manuscript technically sound, and do the data support the conclusions?

Reviewer #2: Yes

3. Has the statistical analysis been performed appropriately and rigorously? 

Reviewer #2: Yes

4. Have the authors made all data underlying the findings in their manuscript fully available?

Reviewer #2: Yes

5. Is the manuscript presented in an intelligible fashion and written in standard English?

Reviewer #2: Yes

6. Review Comments to the Author

Reviewer #2: Thank you for addressing these comments.

Table 2 has been updated adequately. The nesting scores are very helpful to have in the table form and the data from the EPM are explained very well.

7. PLOS authors have the option to publish the peer review history of their article (what does this mean?). If published, this will include your full peer review and any attached files.

Reviewer #2: No

---

## [Editor Report · Acceptance letter]

4 Nov 2021

PONE-D-21-03073R2 

Neuroinflammation, body temperature and behavioural changes in CD1 Male Mice Undergoing Acute Restraint Stress: an Exploratory Study. 

Dear Dr. Calvillo:

I'm pleased to inform you that your manuscript has been deemed suitable for publication in PLOS ONE. Congratulations! Your manuscript is now with our production department. 

Kind regards, 

on behalf of

Dr. Kimberly R. Byrnes 

Academic Editor

PLOS ONE